# Ring finger protein 10 is a novel synaptonuclear messenger encoding activation of NMDA receptors in hippocampus

Margarita C Dinamarca[1†], Francesca Guzzetti[1†], Anna Karpova[2*], Dmitry Lim[3], Nico Mitro[1], Stefano Musardo[1], Manuela Mellone[1], Elena Marcello[1], Jennifer Stanic[1], Tanmoy Samaddar[1], Adeline Burguière[4], Antonio Caldarelli[3], Armando A Genazzani[3], Julie Perroy[4], Laurent Fagni[4], Pier Luigi Canonico[3], Michael R Kreutz[2*], Fabrizio Gardoni[1*], Monica Di Luca[1*]

[1]Dipartimento di Scienze Farmacologiche e Biomolecolari, Università degli Studi di Milano, Milano, Italy; [2]RG Neuroplasticity, Leibniz Institute for Neurobiology, Magdeburg, Germany; [3]Dipartimento di Scienze del Farmaco, Università degli Studi del Piemonte Orientale "Amedeo Avogadro", Novara, Italy; [4]CNRS, Institut de Génomique Fonctionnelle, Montpellier, France

*For correspondence: akarpova@ lin-magdeburg.de (AK); kreutz@ lin-magdeburg.de (MRK); fabrizio. gardoni@unimi.it (FG); monica. diluca@unimi.it (MDL)

[†]These authors contributed equally to this work

Competing interests: The authors declare that no competing interests exist.

**Abstract** Synapses and nuclei are connected by bidirectional communication mechanisms that enable information transfer encoded by macromolecules. Here, we identified RNF10 as a novel synaptonuclear protein messenger. RNF10 is activated by calcium signals at the postsynaptic compartment and elicits discrete changes at the transcriptional level. RNF10 is enriched at the excitatory synapse where it associates with the GluN2A subunit of NMDA receptors (NMDARs). Activation of synaptic GluN2A-containing NMDARs and induction of long term potentiation (LTP) lead to the translocation of RNF10 from dendritic segments and dendritic spines to the nucleus. In particular, we provide evidence for importin-dependent long-distance transport from synapto-dendritic compartments to the nucleus. Notably, RNF10 silencing prevents the maintenance of LTP as well as LTP-dependent structural modifications of dendritic spines.

## Introduction

Understanding how local synaptic events are translated into changes in gene expression is a crucial question in neuroscience. Synapses and nuclei are efficiently connected by bidirectional communication routes that enable transfer of information (*Fainzilber et al., 2011*; *Panayotis et al., 2015*) and regulate the transcription of genes associated with long-term structural changes of neuronal excitability (*Karpova et al., 2012*). NMDAR activation plays a key role in this regard (*Hardingham and Bading, 2010*). NMDARs are heteromeric ionotropic channels that are essential for excitatory neurotransmission (*Paoletti et al., 2013*; *Sanz-Clemente et al., 2013*). NMDARs differ in their subunit composition and, in the forebrain, they can be either di- or tri-heteromeric tetramers consisting of two GluN1 and either one or two GluN2A or GluN2B subunits (*Paoletti et al., 2013*). GluN2A-containing NMDARs are mainly localized at the postsynaptic membrane, at the core of the postsynaptic density (PSD), while the GluN2B-containing NMDARs are also prominently present at extrasynaptic sites (*Hardingham and Bading, 2010*). In addition, numerous reports have indicated that GluN2A- and GluN2B-containing NMDARs may play different roles in the modulation of synaptic plasticity and in central nervous system (CNS) disorders (*Paoletti et al., 2013*). These differences in receptor

**eLife digest** Brain activity depends on the communication between neurons. This process takes place at the junctions between neurons, which are known as synapses, and typically involves one of the cells releasing a chemical messenger that binds to receptors on the other cell. The binding triggers a cascade of events inside the recipient cell, including the production of new receptors and their insertion into the cell membrane. These changes strengthen the synapse and are thought to be one of the ways in which the brain establishes and maintains memories.

However, in order to induce these changes at the synapse, neurons must be able to activate the genes that encode their component parts. These genes are present inside the cell nucleus, which is located some distance away from the synapse. Studies have shown that signals can be sent from the nucleus to the synapse and vice versa, enabling the two parts of the cell to exchange information. Synapses that communicate using a chemical called glutamate have been particularly well studied; but it still remains unclear how the activation of receptors at these "glutamatergic synapses" is linked to activation of genes inside the nucleus at the molecular level.

Dinamarca, Guzzetti et al. have now discovered that this process at glutamatergic synapses involves the movement of a protein messenger to the nucleus. Specifically, activation at synapses of a particularly common subtype of receptor, called NMDA, causes a protein called Ring Finger protein 10 (or RNF10 for short) to move from the synapse to the nucleus. To leave the synapse, RNF10 first has to bind to proteins called importins, which transport RNF10 into the nucleus. Once inside the nucleus, RNF10 binds to another protein that interacts with the DNA to start the production of new synaptic proteins.

Further work is required to identify the molecular mechanisms that trigger RNF10 to leave the synapse. In addition, future studies should evaluate the levels and activity of RNF10 in brain disorders in which synapses are known to function abnormally.

function are probably linked to the cytoplasmic C-tail of the two subunits, which are fairly different and contain specific motifs that bind to PSD-associated scaffolding proteins and to proteins involved in the downstream signal transduction of receptor activation (*Sanz-Clemente et al., 2013*; *Sun et al., 2016*).

It is generally accepted that rises in synaptic, somatic and nuclear $Ca^{2+}$ rapidly regulate gene expression by $Ca^{2+}$-sensing mechanisms (*Bading, 2013*). Recent work showed that following a fast genomic response to sustained rises in $Ca^{2+}$, a considerably slower process that depends on the nuclear import of proteins released from synapses couples local synaptic events to more specific gene expression programs (*Jordan and Kreutz, 2009*; *Ch'ng and Martin, 2011*; *Karpova et al., 2012*). The classical active nuclear import pathway involves the binding of importin α isoforms to nuclear localization signal (NLS) bearing cargo proteins. Interestingly, neuronal importins are present at synapses in association with NMDARs and the PSD (*Dieterich et al., 2008*; *Jeffrey et al., 2009*) and they are able to translocate to the nucleus in response to NMDAR activation (*Thompson et al., 2004*; *Dieterich et al., 2008*). Proteomic studies have identified many proteins in purified synaptosomes that contain a NLS domain, and in the past 10 years, an impressive number of potential synaptonuclear messenger proteins have been characterized (*Ch'ng et al., 2012*; *Karpova et al., 2013*; *Kaushik et al., 2014*). Many of these proteins contain a bona fide NLS and, in some cases, their binding with importin α was shown to be essential for long-distance transport as importin α can serve as an adaptor for nuclear trafficking of cargo in association with a dynein motor (*Karpova et al., 2012*; *Ch'ng et al., 2012*).

In recent studies, it was shown that Jacob, a Caldendrin-binding protein abundantly expressed in limbic brain and cortex (*Dieterich et al., 2008*), encodes and transduces the synaptic and extrasynaptic origin of GluN2B-containing NMDAR signals to the nucleus and elicits the divergent transcriptional responses after activation of these receptors (*Dieterich et al., 2008*; *Karpova et al., 2013*). These findings point to the fascinating possibility that specific NMDAR signals are encoded at synaptic sites and decoded in the nucleus by long-distance trafficking of protein messengers.

Here, we identified Ring Finger protein 10 (RNF10; *Seki et al., 2000*) as a novel synaptonuclear protein messenger, localized at the PSD and specifically associated to the cytoplasmic tail of GluN2A but not GluN2B subunit of NMDARs. After stimulation of synaptic GluN2A-containing NMDARs, RNF10 binds to importin α1 for nuclear long-distance transport. Notably, we show that long-term potentiation (LTP) induces nuclear translocation of RNF10, its interaction with the transcription factor Meox2 and the modulation of gene expression. Most interestingly, RNF10-regulated gene expression appears to feed back to synaptic function.

## Results

### RNF10 is a neuronal synaptic protein

In order to identify new binding partners for GluN2A, we performed a yeast two-hybrid screening using the C-terminal domain (aa 839–1461, without the aa 1462–1464 PDZ-binding sequence) as bait. We obtained a number of positive clones including RNF10 (*Seki et al., 2000*; *Hoshikawa et al., 2008*). In Schwann cells, RNF10 has a function in the transcriptional regulation of myelin formation (*Hoshikawa et al., 2008*). However, very little is known about the neuronal function of this protein (*Seki et al., 2000*; *Lin et al., 2005*; *Hoshikawa et al., 2008*; *Malik et al., 2013*). Using specific glial (GFAP) and neuronal (MAP2) markers, we confirmed that RNF10 is expressed in both glia (*Figure 1A*) and neurons (*Figure 1B*). Interestingly, in neurons RNF10 displayed a nuclear and somatodendritic distribution (*Figure 1B*). Moreover, transfected GFP-RNF10 displayed a prominent co-localization with PSD-95 and GluN2A at the dendritic spines of hippocampal neurons (*Figure 1C*). Similarly, analysis of endogenous RNF10 in *DIV14* primary hippocampal neurons showed clustered RNF10 immunolabeling along dendrites and most puncta co-localized with GluN2A (*Figure 1D*, left panels) and PSD-95 (*Figure 1D*, right panels). PSDs from the rat hippocampus were purified to confirm the subcellular distribution of RNF10 by a biochemical approach. Subcellular fractionation demonstrated that RNF10 is associated with synaptic fractions and that it is prominently present in PSD fractions (*Figure 1E*). Finally, immunofluorescence analysis of the CA1 region of the adult rat hippocampus revealed the presence of an intense signal for RNF10 in the soma and nuclei together with a punctate staining along MAP2-positive dendrites (*Figure 1F*).

Interestingly, GluN2A silencing in primary hippocampal neurons (*Figure 1G*) induced a significant decrease in RNF10 synaptic levels, as indicated by the reduction of RNF10 co-localization with PSD-95 (*Figure 1H*). Notably, the remaining dendritic RNF10 in shGluN2A neurons co-localized with the surface GluN2A pool unaffected by the knock down (*Figure 1I*). However, no modification of RNF10 nuclear level was observed following GluN2A silencing (data not shown; n=30; p=0.5491; shGluN2A vs scramble; unpaired Student's t-test).

### RNF10 interacts with the GluN2A subunit of NMDARs

Different experimental approaches were used to substantiate the yeast two-hybrid data and to confirm the interaction between RNF10 and GluN2A. Co-immunoprecipitation (co-i.p.) studies performed from hippocampal P2 crude membrane fractions indicated a specific interaction of RNF10 with GluN2A but not with GluN2B subunit of the NMDARs (*Figure 2A*). No signal for GluN2A or GluN2B was obtained by using anti-synaptophysin as an irrelevant antibody or in the absence of antibody in the co-i.p. assay (*Figure 2A*). To further validate these findings we performed similar experiments with the GluN2B-associated *synapse-to-nucleus* messenger Jacob (*Dieterich et al., 2008*; *Karpova et al., 2013*). Indeed, the affinity-purified pan-Jacob antibody preferentially co-i.p. GluN2B from rat brain homogenate. Only a very faint band of GluN2A was detected in the complex with Jacob (*Figure 2B*), which might potentially represent synaptic tri-heteromeric GluN1/GluN2A/GluN2B NMDARs.

In addition, cell lysates from COS-7 cells transfected with HA-GluN1 and GFP-GluN2A or GFP-GluN2B constructs were immunoprecipitated with anti-RNF10 and immunoblotted for GFP. The results confirmed that GluN2A interacts with RNF10, whereas GluN2B failed to associate with RNF10 (*Figure 2C*). To corroborate these results, we transfected HA-GluN1 and either GFP-GluN2A or GFP-GluN2B constructs in heterologous cells (COS-7) endogenously expressing RNF10. Immunofluorescence studies revealed the presence of RNF10 aggregates with a high co-localization degree with GluN2A but not with GluN2B (*Figure 2D*; 73.8% ± 2.9% GluN2A/RNF10 vs 31.3.% ± 1.7%

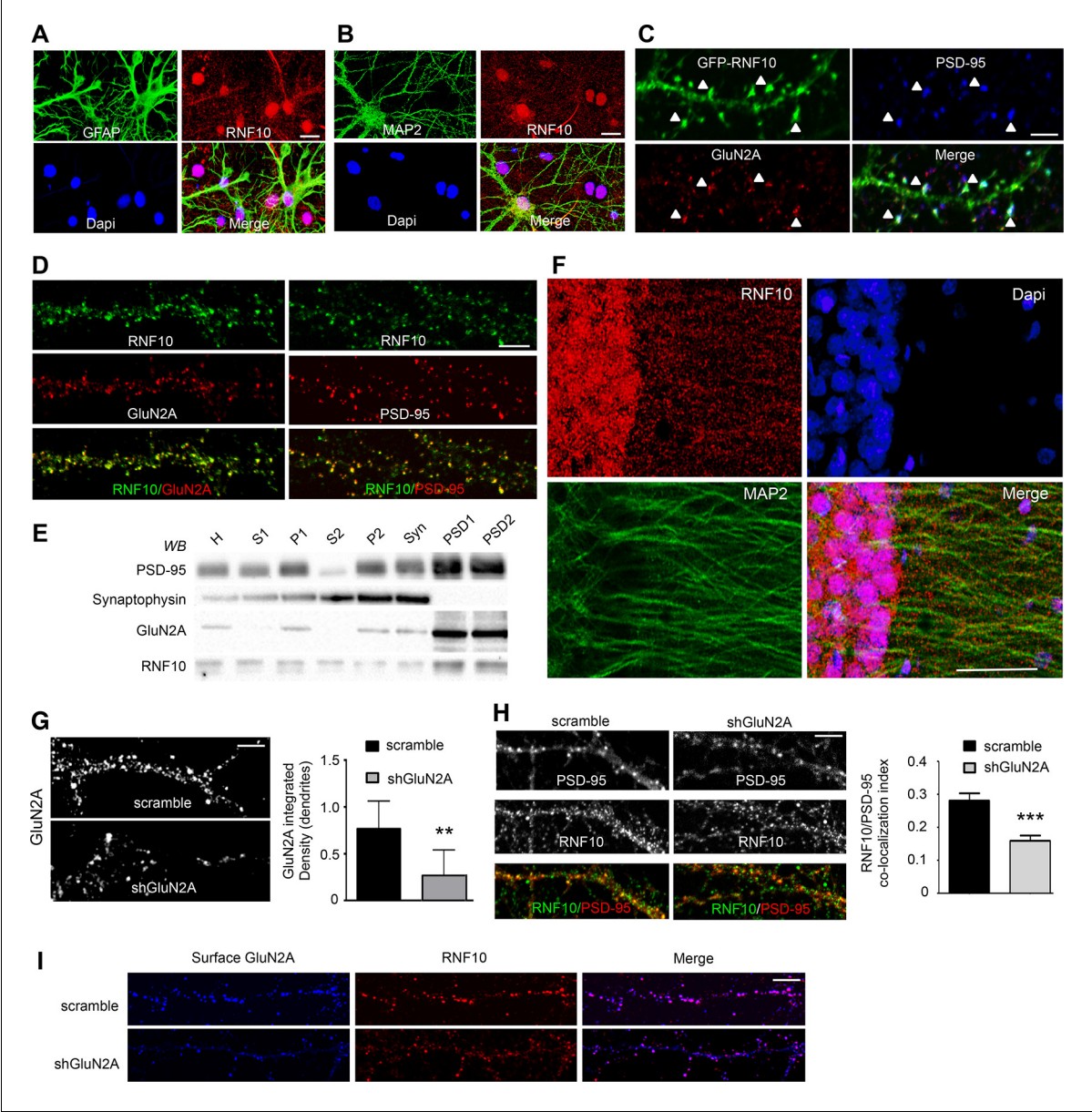

**Figure 1.** RNF10 subcellular distribution in neurons. (**A,B**) Mixed primary hippocampal cultures (*DIV14*) immunolabeled with antibodies for RNF10 (red), glial marker GFAP (A; green) or the neuronal marker MAP2 (B; green), and Dapi (blue) to stain the nucleus; scale bar: 20 µm. (**C**) Dendrite of hippocampal neuron transfected with GFP-RNF10 (*DIV7*) and immunolabeled at *DIV14* for GFP (green), GluN2A (red) and PSD-95 (blue); scale bar: 3 µm. (**D**) High-magnification confocal images of neuronal dendrites (*DIV14*) immunolabeled for endogenous RNF10 (green) and GluN2A (red; left panels) or PSD-95 (red; right panels); scale bar: 4 µm. (**E**) RNF10 and markers of the presynaptic (synaptophysin) and postsynaptic compartment (PSD-95, GluN2A) were analyzed by WB in various subcellular compartments (H: Homogenate fraction; S1: supernatant 1; P1: nuclear fraction; S2: cytosolic fraction 2; P2: crude membrane fraction 2; Syn: synaptosomal fraction; PSD1: Triton Insoluble postsynaptic fraction; PSD2: postsynaptic density fraction). (**F**) Representative confocal images of adult rat hippocampal CA1 pyramidal layer sections showing immunohistochemical labeling for RNF10 (red), MAP2 (green), and Dapi (blue); scale bar: 40 µm. (**G**) Confocal images of dendrites from hippocampal neurons (*DIV14*) transfected at *DIV7* with shGluN2A or scramble vector and immunolabeled for GluN2A; scale bar: 4 µm. The histogram shows the quantification of GluN2A integrated density in dendrites (n=7, **p=0.0069 scramble vs shGluN2A; unpaired Student's t-test). (**H**) GluN2A silencing induces a reduction of RNF10 enrichment at the glutamatergic synapse. Confocal images of primary hippocampal neurons transfected with pGFP-V-RS-scramble (left panels) or with pGFP-V-RS-shGluN2A (right panels) plasmids and immunolabeled (*DIV14*) for RNF10 (green) and PSD-95 (red); scale bar: 4 µm. The histogram shows the quantification of RNF10 co-localization with PSD-95-positive puncta (n=30, ***p<0.001; unpaired Student's t-test). (**I**) Confocal images of dendrites from hippocampal neurons (*DIV14*) transfected at *DIV7* with shGluN2A or scramble vector and immunolabeled for surface GluN2A (blue) and RNF10 (red); scale bar: 4 µm.

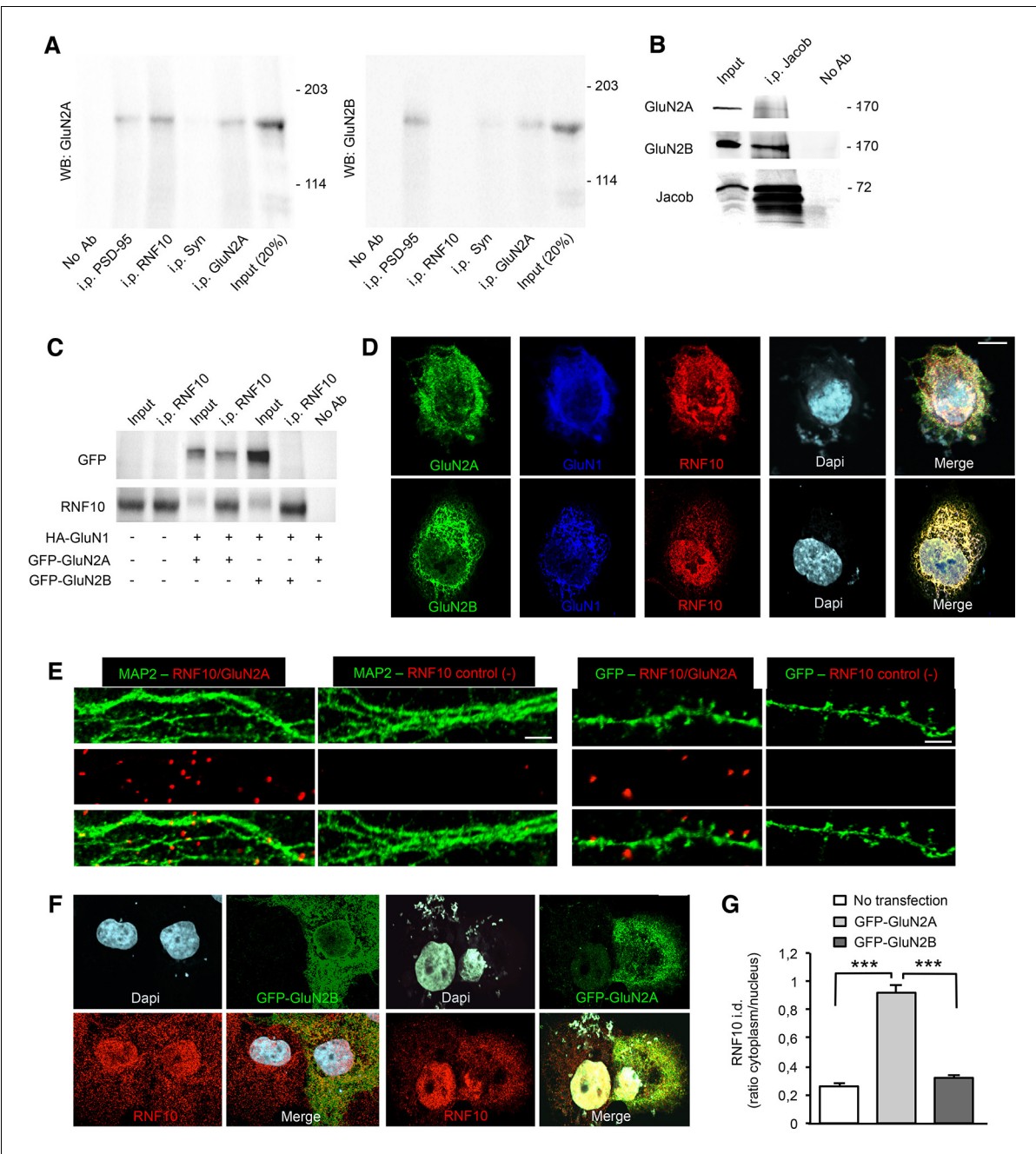

**Figure 2.** RNF10 interaction with GluN2A-containing NMDARs. (**A**) Co-immunoprecipitation (co-i.p.) assay performed in P2 crude membrane fractions by using antibodies against PSD-95, RNF10, synaptophysin (Syn) and GluN2A. WB analysis shows the levels of GluN2A (left panel) and GluN2B (right panel) in the co-immunoprecipitated material. No ab lane: control lane in absence of antibodies during the co-i.p. assay. (**B**) Jacob is a part of the GluN2B receptor complex. Affinity purified pan-Jacob antibodies co-immunoprecipitate GluN2B. (**C**) Co-i.p. assay performed by using an anti-RNF10 antibody from COS-7 cell extracts transfected with HA-GluN1 and GFP-GluN2A or GFP-GluN2B. WB analysis was performed by using anti-GFP and anti-RNF10 antibodies. No ab lane: control lane in absence of antibodies during the co-i.p. assay. (**D**) COS-7 cells expressing RNF10 were transfected with HA-GluN1 and GFP-GluN2A or GFP-GluN2B constructs and immunolabeled for GFP (green), GluN1 (blue), Dapi (cyan) and endogenous RNF10 (red); scale bar: 10 μm. (**E**) In situ detection of proximity between RNF10 and GluN2A (red) along MAP2 (green; left panels) or GFP-positive (green; right panels) dendrites. In control experiments (-), primary hippocampal neurons were labeled with only RNF10 primary antibody and thus only unspecific PLA signals are generated; scale bars: 5 μm (MAP2) and 3 μm (GFP). (**F**, **G**) COS-7 cells expressing RNF10 were transfected with GFP-GluN2A (right panels) or GFP-GluN2B (left panels) constructs and immunolabeled for GFP (green), Dapi (cyan) and endogenous RNF10 (red); scale bar: 10 μm. The histogram shows the quantification of RNF10 integrated density (i.d.) expressed as cytoplasm/nucleus ratio (n=10; ***p<0.001; one-way ANOVA, followed by Bonferroni post-hoc test).

GluN2B/RNF10; p<0.001, n=15; unpaired Student's t-test). Finally, RNF10 clustering with GluN2A at synapses was confirmed by proximity ligation assay (PLA). As shown in *Figure 2E*, a large number of PLA signals were detected when the two antibodies RNF10 and GluN2A were used indicating these two proteins are in close proximity (<40 nm) to each other. As expected, PLA signals were distributed along MAP2-positive dendrites (*Figure 2E*, left panels) and at the top of GFP-positive dendritic spines (*Figure 2E*, right panels). For control experiments, only RNF10 primary antibody was used and no PLA signal was generated (*Figure 2E*).

Notably, transfection of COS-7 cells with GluN2A but not GluN2B induces a highly significant redistribution of endogenous RNF10 from the nucleus to the cytoplasm, thus suggesting that GluN2A traps RNF10 outside the nucleus (*Figure 2F–G*). Overall, these results demonstrate that RNF10 is a component of the excitatory PSD and that it is specifically associated with GluN2A-containing NMDARs.

## The RNF10 N-terminus binds to the juxta-membrane region of GluN2A

We next mapped the binding interface of the interaction between RNF10 and GluN2A using truncation constructs in pull-down and co-i.p. experiments. The pull-down assay performed from adult rat brain tissue using glutathione S-transferase (GST)-GluN2A C-terminal domain fusion proteins (*Figure 3A*) revealed that RNF10 interacts with the GluN2A cytoplasmic tail (CT; aa 839–1464) but failed to bind the distal part of the GluN2A C-terminus (aa 1049–1464) (*Figure 3B*, upper panel). In addition, GluN2A(839–991) failed to interact with RNF10 (*Figure 3B*, lower panel), thus indicating GluN2A(991–1049) is needed for the formation of the GluN2A/RNF10 complex.

To strengthen these results, GFP-GluN2A (1–1049) truncation mutant (bearing a stop codon at aa 1049) was transfected in COS-7 cells to evaluate its capability to interact with endogenous RNF10 (*Figure 3C,D*). Immunofluorescence assay revealed a high GFP-GluN2A(1–1049)/RNF10 clustering leading also to RNF10 redistribution from the nucleus to the cytoplasm (*Figure 3C*; cytoplasm/nucleus ratio 0.98 ± 0.04 vs 0.26 ± 0.02 GFP-GluN2A(1–1049) vs untransfected, p<0.001, n=10; unpaired Student's t-test), similarly to what has been described above for GFP-GluN2A construct (see *Figure 2*). Accordingly, RNF10 co-immunoprecipitated with both GFP-GluN2A and GFP-GluN2A (1–1049) from COS-7 lysates (*Figure 3D*), confirming that RNF10 interacts with the GluN2A juxta-membrane aa 839–1049 region. Interestingly, GluN2A(991–1029) domain previously shown to be responsible for a calcium-dependent binding of the NMDAR subunit with Calmodulin (CaM; *Bajaj et al., 2014*). Based on this consideration, we first evaluated whether calcium ($Ca^{2+}$) could modulate GluN2A/RNF10 interaction. Using a pull-down assay we found that the presence of free $Ca^{2+}$ (2 mM) in the buffer significantly reduced GluN2A binding to GST-RNF10 FL (*Figure 3E*; -37.8% ± 6.2%; p<0.01). Similarly presence of free $Ca^{2+}$ (2 mM) in the buffer significantly reduced RNF10 binding to GST-GluN2A(839–1464) (*Figure 3F*; -65.4 ± 8.8%; p<0.05). As previously reported (*Bajaj et al., 2014*), $Ca^{2+}$ induced also the binding of CaM to the GluN2A fusion protein (*Figure 3F*). In addition, co-incubation with exogenous CaM completely prevented the interaction between RNF10 and GluN2A in the pull-down assay (*Figure 3G*) only in the presence of $Ca^{2+}$ in the buffer (*Bajaj et al., 2014*). Similarly, a co-i.p. assay from COS-7 cells transfected with GFP, GFP-GluN2A or GFP-GluN2B confirmed the capability of $Ca^{2+}$/CaM to disrupt RNF10/GluN2A complex (*Figure 3H*).

The RNF10 protein contains a binding sequence for the transcription factor Mesenchyme Homeobox 2 (Meox2; Meox2 Binding Domain, MBD, aa 101–185; *Lin et al., 2005*), a Ringer Finger Domain (RFD, aa 225–270) and two putative nuclear localization sequences (NLS1, aa 591–599 and NLS2, aa 784–791). Several Myc-RNF10 mutants were prepared and co-transfected with GluN2A in COS-7 cells in order to identify the RNF10 domain responsible for the interaction with the NMDAR subunit. The RNF10(221–802) construct failed to interact with GluN2A as demonstrated by both co-i.p. (*Figure 3I*) and co-localization (*Figure 3K,L*) assays when compared to RNF10 full length (FL). Conversely, all RNF10 truncation mutants bearing the RNF10(1–221) domain co-immunoprecipitated with GFP-GluN2A (*Figure 3J*). Finally, pull-down assay performed from adult rat brain tissue using GST-RNF10 fusion proteins confirmed that the RNF10 N-terminal region 1–221 is crucial for the binding to GluN2A (*Figure 3M*).

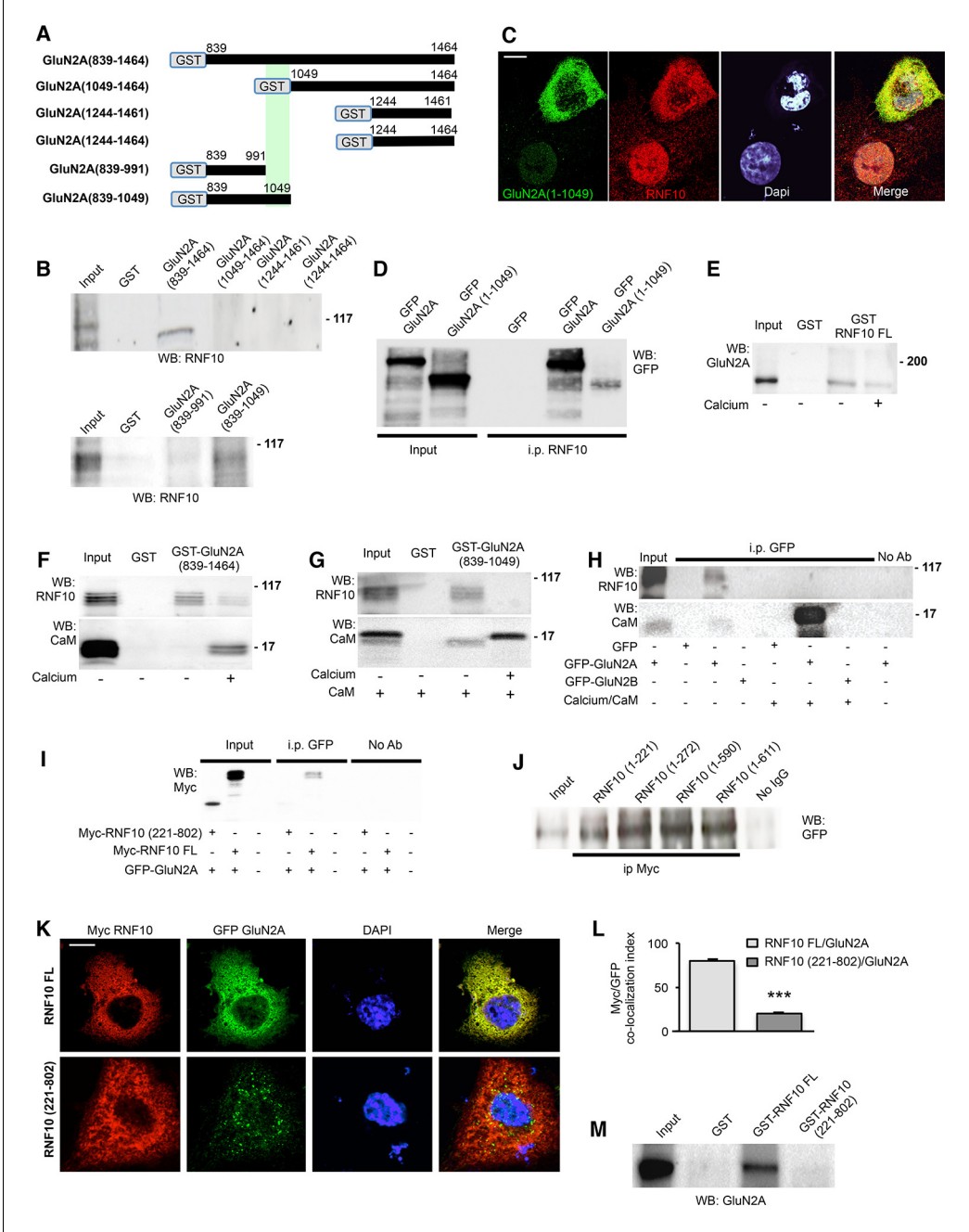

**Figure 3.** RNF10 N-terminal domain interacts with the juxtamembrane region of GluN2A C-tail. (**A**) Scheme showing GST-GluN2A fusion proteins used in the pull-down assay. (**B**) GST and GST-GluN2A fusion proteins were incubated in a pull-down assay with rat hippocampal extracts. WB analysis was performed with RNF10 antibody. (**C**) Confocal images of COS-7 cells transfected with GFP-GluN2A (1–1049) and immunolabeled for GFP (green), Dapi (cyan) and RNF10 (red); scale bar: 10 μm. (**D**) Co-i.p. assay performed from lysates of COS-7 cells transfected with GFP-GluN2A or GFP-GluN2A(1–1049). WB analysis was performed by using a GFP antibody (JL-8). (**E**) GST and GST-RNF10 full-length (FL) fusion proteins were incubated in a pull-down assay with rat hippocampal extracts in presence or absence of calcium (2 mM). WB analysis was performed with GluN2A antibody. (**F**) GST and GST-GluN2A(839–1464) fusion proteins were incubated in a pull-down assay with rat hippocampal extracts in presence or absence of calcium (2 mM). WB analysis was performed with RNF10 and CaM antibodies. (**G**) GST and GST-GluN2A(839–1049) fusion proteins were incubated in a pull-down assay with rat hippocampal extracts with CaM (0.1 μM) in the presence or absence of calcium (2 mM). WB analysis was performed with RNF10 and CaM antibodies. (**H**) Co-i.p. assay performed from lysates of COS-7 cells transfected with GFP, GFP-GluN2A or GFP-

*Figure 3 continued on next page*

*Figure 3 continued*

GluN2B in the presence or absence of calcium (2 mM)/CaM (0.1 µM). WB analysis was performed by using RNF10 and CaM antibodies. (I) Co-i.p assay performed by using a GFP antibody from lysates of COS-7 cells transfected with RNF10 FL, RNF10(221–802) and GFP-GluN2A. WB analysis was performed by using a Myc antibody. No ab lanes: control lanes in absence of antibodies during the co-i.p. assay. (J) Co-i.p assay performed by using a Myc antibody from lysates of COS-7 cells transfected with RNF10 truncation mutants and GFP-GluN2A. WB analysis was performed by using a GFP antibody (JL-8). (K,L) COS-7 cells expressing RNF10 were transfected with GFP-GluN2A and Myc-RNF10 FL or Myc-RNF10 (221–802) constructs and immunolabeled for GFP (green), Myc (red) and Dapi (blue) (G); scale bar: 10 µm. The histogram (H) shows the quantification of Myc/EGFP co-localization index [n=10; ***p<0.001 RNF10 FL vs RNF10(221–802); unpaired Student's t-test]. (M) GST and GST-RNF10 fusion proteins were incubated in a pull-down assay with rat hippocampal extracts. WB analysis was performed with GluN2A antibody.

## RNF10 silencing induces molecular and morphological modifications of the glutamatergic synapse

RNF10 is a member of the Ring Finger Protein family, which has been generally implicated in development, transcriptional regulation, signal transduction, DNA repair and oncogenesis (*Saurin et al., 1996*). However, the neuronal function of RNF10 is still unknown (*Lin et al., 2005*; *Malik et al., 2013*). To understand the role of RNF10 in neurons, we silenced RNF10 expression by using a short hairpin (sh) RNF10 knock-down or a scramble sequence (as control) in primary hippocampal cultures. We tested three different sequences of RNF10 shRNA (shRNF10; see Materials and methods); shRNF10 that led to the highest level of RNF10 downregulation (>90%; TRCN0000041128) was selected and used in all experiments. Confocal imaging of hippocampal neurons transfected with shRNF10 or scramble plasmids demonstrated a significant effect of RNF10 knock-down on dendritic spine morphology (*Figure 4A–E*). In particular, RNF10 silencing produced a significant reduction in dendritic spine density (*Figure 4B*) without any effect on dendritic spine length (*Figure 4C*) or dendritic spine head width (*Figure 4D*). For a more detailed morphological analysis, dendritic spines were categorized according to their shape (mushroom, thin and stubby) using a validated classification method (*Bourne and Harris, 2008*). However, no effect of shRNF10 in the proportion of dendritic spine subtypes was observed (*Figure 4E*). Notably, the effect of RNF10 silencing on dendritic spine density was fully rescued by co-expressing a wild-type human variant of RNF10 resistant to shRNA (see Materials and methods; flag-RNF10; *Figure 4A–E*). Interestingly, overexpression of RNF10 per se had no effect on dendritic spine morphology (data not shown). We then verified whether the loss of spines following RNF10 protein knock-down was correlated with an altered expression of the main components of the excitatory synapse. To this end, we infected primary hippocampal neurons (*Figure 4F*, left panels) and organotypic hippocampal slices (*Figure 4F*, right panels) with pLKO-shRNF10 lentivirus or scramble sequence as a control. As expected, RNF10 silencing produced a significant reduction of RNF10 protein level compared to dissociated neurons and slices treated with scramble construct (*Figure 4F*). Most importantly, RNF10 silencing resulted in a significant decrease of GluN2A, PSD-95 and the GluA1 subunit of the AMPA receptor (AMPAR) protein levels in the total cell homogenate of pLKO-shRNF10-infected dissociated neurons and organotypic slices (*Figure 4F*).

To learn more about the effect of RNF10 knock-down on gene expression, we performed a microarray analysis from organotypic hippocampal slices virally infected with pLKO-shRNF10 lentivirus in order to identify RNF10 target genes. Interestingly, heat map of differentially expressed genes (*Figure 5A*), gene ontology analysis (*Figure 5B*) and real-time PCR validation (*Figure 4G*) showed that RNF10 silencing modulated the expression of several genes involved in excitatory synaptic transmission and dendritic spine morphology (*Vogt et al., 2007*; *Michaluk et al., 2011*; *Ramakers et al., 2012*). Furthermore, we confirmed by WB analysis the effect of viral infection with pLKO-shRNF10 on protein levels of some of the RNF10 target genes, such as Ophn1, ArhGap4 and ArhGef6 (*Figure 4H*).

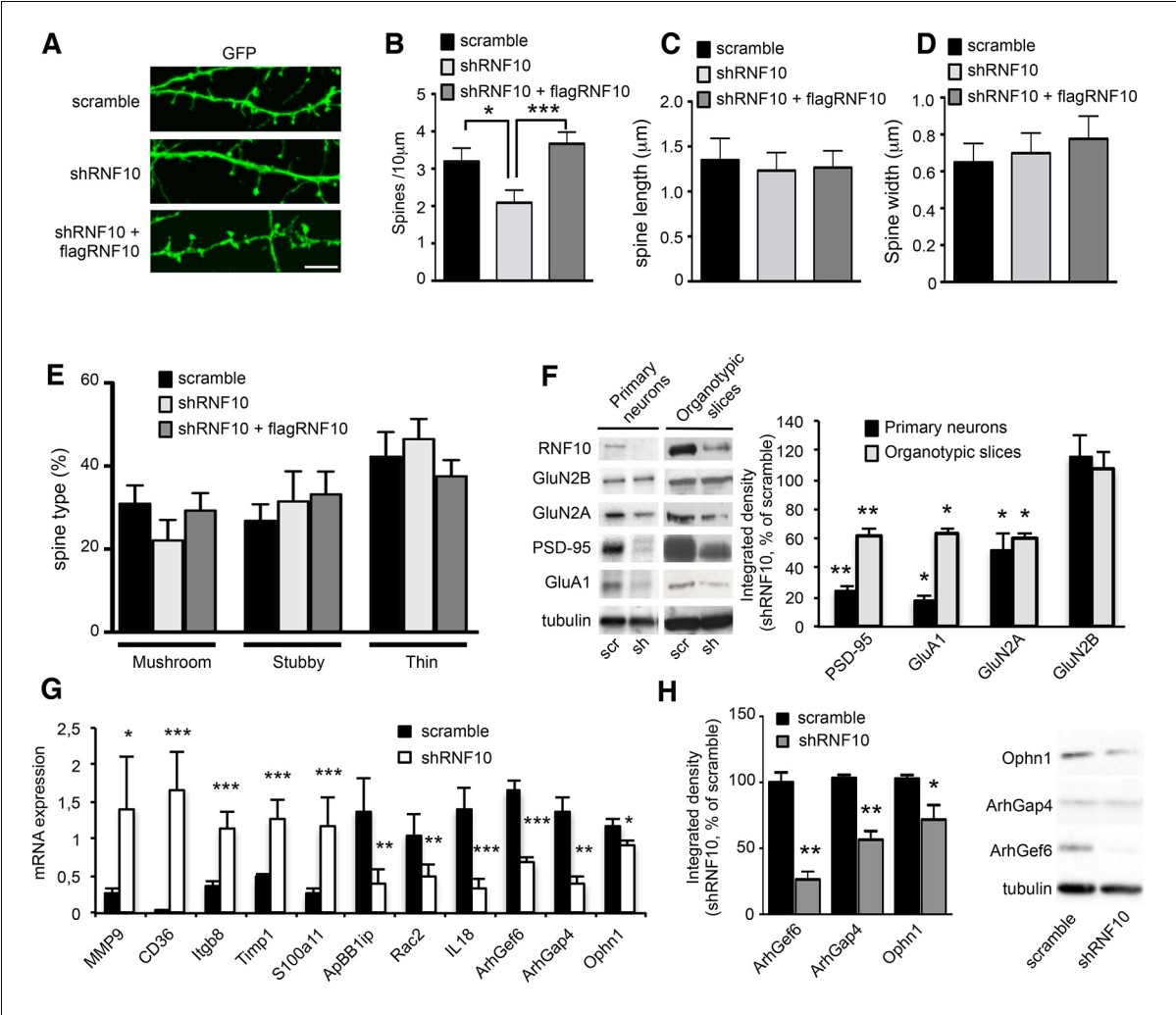

**Figure 4.** RNF10 silencing induces molecular and morphological modifications of the glutamatergic synapse. (A) Confocal images of primary hippocampal neurons (*DIV14*) transfected at *DIV7* with pGIPZ-scramble, shRNF10 and shRNF10 plus flagRNF10 and immunolabeled for GFP (green); scale bar: 5 μm. (B-E) Histograms showing the quantification of dendritic spine density (B) (n=6–10; *p<0.05, scramble vs shRNF10; ***p<0.001, shRNF10 vs shRNF10 + flagRNF10; one-way ANOVA, followed by Tukey post-hoc test), dendritic spine length (C), dendritic spine head width (D) and dendritic spine type (E). (F) WB analysis from homogenates of primary hippocampal neurons (*DIV14*) and organotypic hippocampal slices (*DIV14*) lentivirally infected with pGIPZ-scramble sequence (scramble) as control or with pLKO-shRNF10 (shRNF10). The histogram shows the quantification of the expression levels of GluN2A, GluA1, PSD-95 and GluN2B in shRNF10-infected neurons and slices, normalized on tubulin and expressed as % of scramble (n=6; *p<0.05; **p<0.01; unpaired Student's t-test). (G) mRNA expression levels of genes associated with synaptic transmission or dendritic spine morphology by real-time PCR from *DIV14* organotypic hippocampal slices lentivirally infected (*DIV4*) with pGIPZ-scramble sequence (scramble) as control or with pLKO-shRNF10 (shRNF10) (n=4, ***p<0.001; **p<0.01; *p<0.05; unpaired Student's t-test). (H) WB for ArhGef6, ArhGap4, Ophn1 and tubulin from cell lysates of organotypic hippocampal slices infected with pGIPZ-scramble or with pLKO-shRNF10. The histogram shows the quantification of protein levels from shRNF10 samples with respect to pGIPZ-scramble, following normalization on tubulin (n=3, *p<0.05; **p<0.01; unpaired Student's t-test).

## Neuronal activity regulates RNF10 synaptonuclear localization

The synaptic and nuclear localization in neurons, the presence of two different NLS motifs and its interaction with GluN2A and Meox2 (*Lin et al., 2005*) suggest that RNF10 could be a novel synapto-nuclear protein messenger. To validate this hypothesis, we assessed whether the modulation of neuronal activity affects the subcellular localization of RNF10 and its association with the interacting proteins. We first enhanced synaptic excitatory activity with the GABA-A receptor antagonist Bicu-culline (50 μM) in the presence of the K⁺ channel blocker 4-AP (2.5 mM; *Hardingham et al., 2002*) ('Bic' treatment). Enhanced excitatory activity significantly decreased RNF10 immunoreactivity along

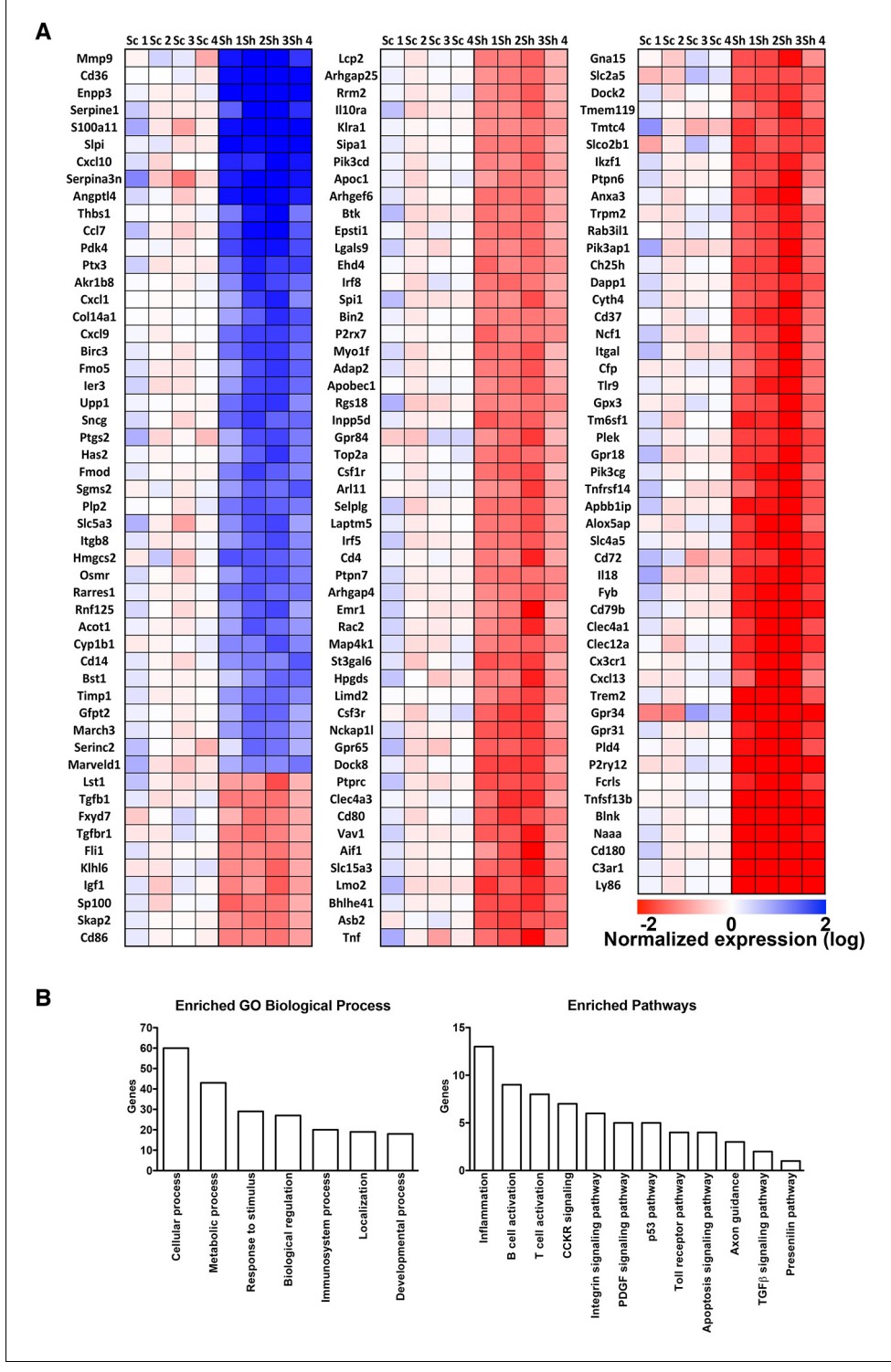

**Figure 5.** Heat map and gene ontology of differentially expressed genes identified in microarray experiments. (**A**) Heat map of differentially expressed genes. Expression data are reported as log2 and blue color indicates high expression values and red color low expression value. Sc represents the different scramble control (1–4), while Sh represents the shRNA against RNF10 (5–8). (**B**) Gene ontology analysis of biological processes and enriched pathways analysis of differentially regulated genes in absence of RNF10.

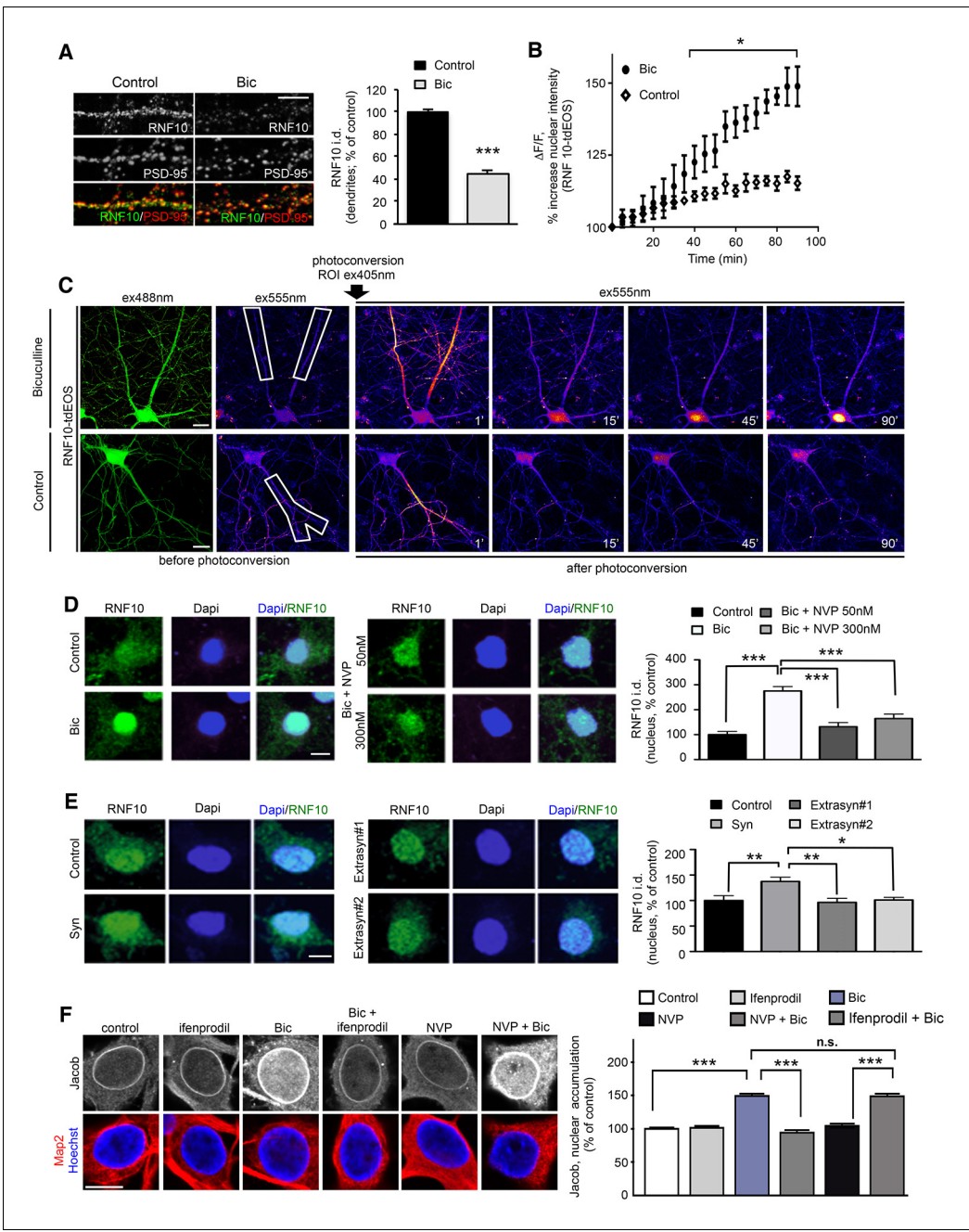

**Figure 6.** Activation of synaptic NMDARs triggers RNF10 translocation to the nucleus. (**A**) Hippocampal neurons (*DIV14*) were incubated for 24 hr with 50 μM Bicuculline and 2.5 mM 4-AP ('Bic') and immunolabeled for RNF10 (green) and PSD-95 (red). The histogram shows the quantification of RNF10 levels along dendrites 24 hr after treatment (n=30, ***p<0.001; unpaired Student's t-test); scale bar: 4 μm. (**B,C**) Bic treatment induces RNF10-tdEOS translocation from distal dendrites to the nucleus in hippocampal neurons. (**B**) The histogram shows a significant increase in RNF10-tdEOS photoconverted fluorescent intensities in the nucleus following Bic treatment (n=6; p<0.05 Bic vs. control, from 40' to 90'; unpaired Student's t-test). (**C**), left panels: Baseline confocal image of RNF10-tdEOS expressing hippocampal neuron illuminated sequentially with 488 nm and 555 nm laser excitation wavelengths showing no emitted signal in the red spectra (ex555nm; left panels). Distal dendrite (ROI) selected for photoconversion was illuminated with UV laser (405 nm wavelengths) repetitively through the image z-stack. (**C**), right panels: Depicted are confocal max intensity projection images at respective time points in control after Bic treatment or in control (untreated) neurons; scale bar: 20 μm. (**D**) Hippocampal neurons (*DIV14*) were treated with Bic in presence of the GluN2A inhibitor NVP-AAM007 at different concentrations (50 and 300 nM), immunolabeled for RNF10 (green) and stained with Dapi (blue). The histogram shows the quantification of RNF10 integrated

*Figure 6 continued on next page*

*Figure 6 continued*

density in the nucleus expressed as % of control neurons (n=10, ***p<0.001 control vs Bic, Bic vs Bic+NVP 50 nM and Bic vs Bic+NVP 300 nM; one-way ANOVA followed by Bonferroni post-hoc test); scale bar: 10 μm. (**E**) Hippocampal neurons (*DIV14*) treated with 'Syn' (50 μM Bicuculline; 2.5 mM 4-AP; 5 μM Ifenprodil, 8 hr), with 'Extrasyn#1' or 'Extrasyn#2' protocols (see Materials and methods), immunolabeled for RNF10 (green) and stained with Dapi (blue). Histogram showing the quantification of RNF10 integrated density in the nucleus expressed as % of control neurons (n=8, *p<0.05 Extrasyn#2 vs Syn, **p<0.01 Syn vs Extrasyn#1 and Syn vs. control; one-way ANOVA followed by Bonferroni post-hoc test); scale bar: 10 μm. (**F**) Depicted are representative laserscans averaged from three confocal sections of the nucleus of *DIV16* hippocampal primary neurons immunolabeled with affinity purified antibodies against pan-Jacob (rabbit) and co-labeled with anti-MAP2 antibodies as a neuronal specific marker. Neuronal nuclei are outlined with the DNA stain Hoechst 34580. Scale bar: 10 μm. Relative fluorescence intensities of Jacob 30 min of synaptic stimulation with and without selective inhibitors were normalized to untreated non-stimulated control (n=31–70, ***p<0.001; one-way ANOVA followed by Bonferroni post-hoc test).

dendrites as compared to untreated neurons, without affecting PSD-95 (*Figure 6A*) labeling and induced a significant increase in RNF10 nuclear staining (*Figure 6D*, left panels). This effect was not correlated to an alteration of RNF10 total protein amount as determined by WB analysis performed with neuronal lysates (data not shown; +7.8% ± 8.6%, cLTP vs. control; p>0.05, n=6). To further confirm that RNF10 accumulates in the nucleus following long-distance protein transport, we performed time-lapse confocal imaging of RNF10 fused to photoconvertible tdEOS (*Figure 6B,C*). We photoconverted RNF10-tdEOS in distal dendrites from green-to-red emission using 405 nm irradiation immediately after Bic treatment and tracked the red fluorescence signal over 90 min. Bic treatment induced a significant increase in RNF10-tdEOS photoconverted fluorescence intensity in the nucleus compared to untreated neurons accompanied by a decline in distal dendrites (*Figure 6B–C*).

Interestingly, the co-incubation of Bic with the GluN2A-specific inhibitor NVP-AAM007 (NVP, 50 nM or 300 nM) led to a significant reduction of RNF10 nuclear accumulation compared to Bic treatment alone also at a concentration of 50 nM, which is known to preferentially block di-heteromeric GluN1/GluN2A and not tri-heteromeric GluN1/GluN2A/GluN2B NMDARs (*Figure 6D*; *Foster et al., 2010*). Similarly, Bic dependent increase in RNF10 nuclear staining was reduced by the co-incubation with the NMDAR blocker MK801 (data not shown). Moreover, hippocampal neurons (*DIV14*) were treated with Bicuculline (50 μM) and 4-AP (2.5 mM) in presence of the GluN2B antagonist ifenprodil (5 μM), corresponding to a well-validated protocol to obtain a specific activation of synaptic NMDARs (Syn; *Figure 6E*, left panel; *Hardingham et al., 2002*). Synaptic NMDARs stimulation induced a significant increase in RNF10 nuclear accumulation (*Figure 6E*, left panels). On the other hand, two different protocols known to activate extrasynaptic NMDARs (*Hardingham et al., 2002*; *Karpova et al., 2013*) failed to induce the translocation of RNF10 from dendrites to the nucleus (*Figure 6E*, right panel). In control experiments, we found that enhanced synaptic activity with Bicuculline (50 μm) and 4-AP (2.5 mM) resulted in an increase in nuclear Jacob as reported previously (*Figure 6F*; *Behnisch et al., 2011*; *Karpova et al., 2013*). This increase was abolished in the presence of the GluN2B antagonist ifenprodil (5 μM), but not with GluN2A antagonist NVP-AAM077 (50 nM) indicating that Jacob dissociates from the synapses following activation of GluN2B but not di-heteromeric GluN2A containing NMDAR (*Figure 6F*).

## Neuronal activity modulates RNF10 interaction with protein partners

The import of proteins from the cytosol into the nucleus through the nuclear pore complex depends on the binding of importins to a specific nuclear localization sequence (NLS; *Thompson et al., 2004*; *Jordan and Kreutz, 2009*; *Karpova et al., 2012*). According to this scheme, importins function as adapter molecules by binding NLS-bearing proteins. Components of the classical nuclear import machinery are present at synapses in association with NMDARs and importins translocate to the nucleus in response to NMDAR stimulation (*Thompson et al., 2004*; *Jeffrey et al., 2009*; *Marfori et al., 2011*; *Ch'ng et al., 2012*). Analysis of the RNF10 sequence showed the presence of two putative NLS motifs (NLS1: aa 591–599; NLS2: aa 784–791; *Seki et al., 2000*; *Marfori et al., 2011*). We therefore investigated the neuronal localization of RNF10 by transfection of RNF10 FL and RNF10 truncated mutants lacking one [Myc-RNF10 (1–611)] or both NLS motifs [Myc-RNF10 (1–

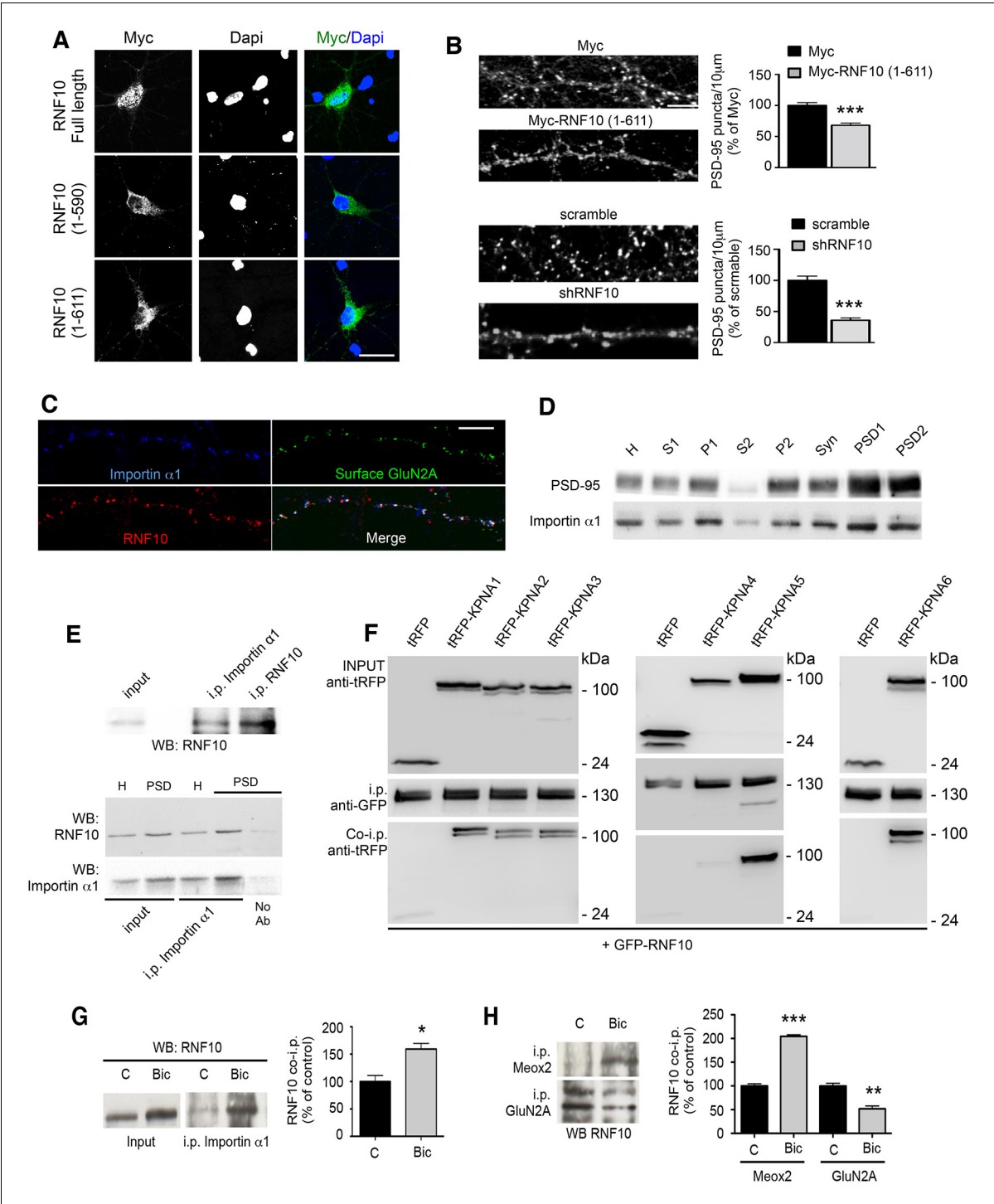

**Figure 7.** RNF10 translocates into the nucleus through NLS2-dependent interaction with importin α1. (**A**) Confocal images of primary hippocampal neurons transfected with Myc-RNF10 full-length (FL) or Myc-RNF10 truncated mutants and immunolabeled for Myc (green) and Dapi (blue); scale bar: 20 μm. (**B**) Primary hippocampal neurons transfected with Myc, Myc-RNF10 (1–611), pGIPZ-scramble or with shRNF10 were immunolabeled (*DIV14*) for PSD-95. The histograms show the quantification of PSD-95-positive puncta expressed as % of Myc (upper histogram) or scramble (lower histogram) transfected neurons (n=50–88, ***p<0.001, Myc-RNF10 (1–611) vs Myc and shRNF10 vs scramble; unpaired Student's t-test); scale bar: 4 μm. (**C**) Confocal images of dendrites of hippocampal neurons (*DIV14*) immunolabeled for RNF10 (red), surface GluN2A (green) and importin α1 (blue); scale bar: 4 μm. (**D**) Importin α1 and PSD-95 protein levels were analyzed by means of WB analysis in various subcellular compartments purified from rat hippocampal tissue (H: Homogenate fraction, S1: supernatant 1, P1: nuclear fraction, S2: cytosolic fraction 2, P2: crude membrane fraction 2, Syn: synaptosomal fraction, PSD1: Triton insoluble postsynaptic fraction, PSD2: postsynaptic density fraction). (**E**) Representative co-i.p. assay showing the

*Figure 7 continued on next page*

*Figure 7 continued*

interaction between RNF10 and importin α1 in hippocampal tissue homogenate (upper and lower panel) and PSD fraction (lower panel). No Ab lane: control lane in absence of the antibody. WB analysis was performed with RNF10 and importin α1 antibodies. (**F**) RNF10-tagged with GFP was co-expressed with multiple importin α isoforms (*KPNA1-KPNA6*) tagged with tagRFP in HEK293T cells. RNF10-GFP was immunoprecipitated from cell extract using anti-GFP MicroBeads. Co-immunoprecipitated importinerase - α isoforms were detected in complex with RNF10 using anti-tagRFP antibodies. (**G**) Co-i.p. assay performed by using an importin α1 antibody from cell homogenates of control (C) neurons or treated with Bic. WB analysis was performed with RNF10 antibody. The histogram shows the quantification of RNF10/importin α1 interaction expressed as % of control (n=3, *p<0.05; unpaired Student's t-test). (**H**) Representative co-i.p. assay from cell homogenates of control (C) neurons or treated with Bic. WB analysis was performed with RNF10 antibody. The histogram shows the quantification of RNF10 interaction with GluN2A and Meox2 expressed as % of control (n=3, ***p<0.001, Meox2, Bic vs control; **p<0.01, GluN2A, Bic vs control; unpaired Student's t-test).

590)] in primary hippocampal neurons. Myc-RNF10 FL showed an intense nuclear accumulation (as the endogenous protein; *Figure 7A*). However, both truncated mutants failed to localize to the nucleus as indicated by a very low co-localization with the nuclear marker Dapi (*Figure 7A*). In particular, also the longest Myc-RNF10 (1–611) construct lacking only the C-terminal NLS2 domain accumulated in the cell soma but did not show a marked nuclear localization (*Figure 7A*), thus suggesting that the second NLS2 motif could be responsible for RNF10 nuclear translocation in neurons. Notably, the overexpression of Myc-RNF10 (1–611) construct produced a significant reduction in the density of PSD-95-positive puncta along dendrites (*Figure 7B*, upper panels) similar to that observed following RNF10 silencing (*Figure 7B*, lower panels), indicating that this protein fragment can act in a dominant negative manner.

Based on these results, we next evaluated a possible RNF10 interaction with importins in hippocampal neurons. RNF10 displayed a prominent co-localization with importin α1 (*KPNA2* gene product; see *Zienkiewicz et al., 2013*) and surface GluN2A along dendrites of *DIV14* primary hippocampal neurons (*Figure 7C*). In agreement with previous data about other importins (*Thompson et al., 2004*; *Jeffrey et al., 2009*; *Marfori et al., 2011*; *Ch'ng et al., 2012*), analysis of importin α1 subcellular distribution showed its presence in purified PSDs from rat hippocampus (*Figure 7D*). Immunoprecipitation experiments performed revealed that RNF10 interacts with importin α1 in neuronal cells (*Figure 7E*, upper panel). Interestingly, RNF10 co-precipitation with importin α1 was detected both in PSD and homogenate fraction but it was much more prominent in PSDs (*Figure 7E*, lower panel). In order to verify the capability of RNF10 to interact also with other importins, we performed heterologous co-immunoprecipitation experiments of RNF10 fused to GFP with six importin α isoforms (*KPNA1-KPNA6* gene products) fused to tag-RFP. As shown in *Figure 7F*, RNF10 can interact with majority of importin α isoforms except importin α3 (*KPNA4* gene product).

Bic treatment significantly increased RNF10/importin α1 association (*Figure 7G*). Interestingly, also the RNF10 interaction with the transcription factor Meox2 was significantly increased by Bic treatment (*Figure 7H*). On the contrary, Bic treatment significantly decreased RNF10/GluN2A interaction compared to control neurons (*Figure 7H*), which is in agreement with the reduction of RNF10 labeling along dendrites (see *Figure 6A*).

Taken together these findings indicate that enhancing synaptic activity of di-heteromeric GluN2A-containing NMDARs results in the dissociation of RNF10 from the cytoplasmic tail of GluN2A at postsynaptic sites and a tighter association with neuronal importin and importin-dependent nuclear translocation, followed by an association of RNF10 with the transcription factor Meox2.

## RNF10 nuclear translocation is mediated by LTP induction

LTP is widely considered as one of the major cellular mechanisms that underlie learning and memory (*Bliss and Collingridge, 1993*) and requires de novo protein synthesis and gene transcription for its maintenance (*Smolen et al., 2012*). We next asked whether RNF10 translocates to the nucleus from distal dendrites following induction of LTP in mature hippocampal neuronal cultures. For induction of LTP we used two established protocols (*Deisseroth et al., 1996*; *Otmakhov et al., 2004*; *Oh et al., 2006*). First, we employed 18s@50Hz field stimulation, which is known to produce a stable synaptic potentiation in cultured hippocampal neurons (*Deisseroth et al., 1996*) and in parallel we performed time-lapse confocal imaging of RNF10 fused to photoconvertible tdEOS (*Figure 8A–C*). The rise in intracellular Ca$^{2+}$ concentration upon LTP-inducing stimuli was verified by Fluo-4-AM fluorescence (data not shown). We photoconverted RNF10-tdEOS in distal dendrites (*Figure 8A*) and

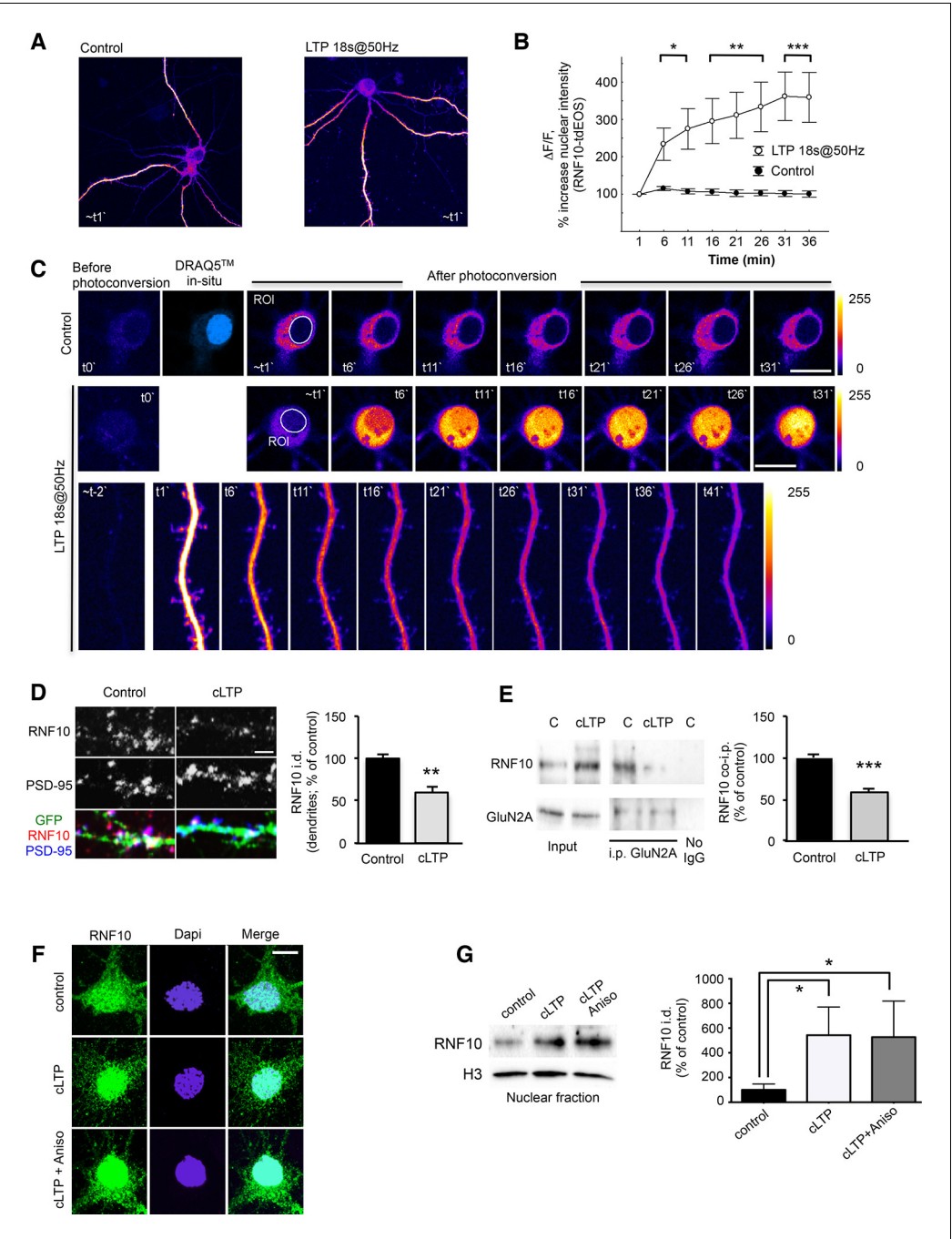

**Figure 8.** LTP induction triggers RNF10 translocation to the nucleus. (**A–C**) RNF10-tdEOS translocates from distal dendrites to the nucleus in mature hippocampal neurons upon high-frequency 18s@50Hz field synaptic stimulation. Distal dendrites (ROIs) selected for photoconversion were illuminated with UV laser (405 nm wavelengths) repetitively through the image z-stack (image t0'). (**A**) Representative images of control and stimulated RNF10-tdEOS expressing hippocampal neurons illuminated with 568 nm laser excitation wavelength 1 min following photoconversion. (**B**) The histogram shows a significant increase in RNF10-tdEOS photoconverted fluorescent intensities in the nucleus following high frequency 18s@50Hz field synaptic stimulation (n=9, *p<0.05, **p<0.01, ***p<0.001; unpaired Student's t-test). (**C**) Depicted are confocal max intensity projection images of control and stimulated RNF10-tdEOS expressing hippocampal neurons at respective time points after stimulation. Lower panels: magnified image sequence of UV illuminated dendritic segment with multiple spines is represented showing the decrease in distal dendrites upon field stimulation. All experiments were performed in a presence of anisomycin (7.5 μM). (**D**) Confocal images of dendrites of hippocampal neurons (*DIV14*) transfected with GFP

*Figure 8 continued on next page*

*Figure 8 continued*

(green) to visualize neurites and immunolabeled for RNF10 (red) and PSD-95 (blue). The histogram shows the quantification of RNF10 signal in dendrites 2 hr after induction of cLTP (see Materials and methods) expressed as % of control (n=5, **p<0.01; unpaired Student's t-test); scale bar: 4 µm. (E) Co-i.p. assay from hippocampal extracts performed with an antibody against GluN2A. WB analysis was performed using antibody for RNF10 and GluN2A. The graph shows the effect of cLTP induction on RNF10 interaction with GluN2A expressed as % of control (n=4, ***p<0.001; unpaired Student's t-test). (F) Representative confocal images of hippocampal neurons (*DIV14*) after the induction of cLTP in the presence or absence of 7.5 µM anisomycin and immunolabeled for RNF10 (green) and Dapi (blue); scale bar: 10 µm. (G) WB analysis for RNF10 from P1 nuclear fraction purified from hippocampal neurons after the induction of cLTP in the presence or absence of 7.5 µM anisomycin. The histogram shows the quantification of RNF10 integrated density normalized on Histone-H3 (n=4, *p<0.05, control vs cLTP and control vs cLTP+Anisomycin; one-way ANOVA, followed by Tukey post-hoc test).

tracked the red fluorescence signal over time. We observed a significant increase in RNF10-tdEOS photoconverted green-to-red fluorescence intensity in the nucleus when compared to unstimulated neurons (*Figure 8B,C*) accompanied by a concomitant decline in distal dendrites upon stimulation (*Figure 8C*, lower panels). This emphasizes that the synaptodendritic but not the somatic pool of RNF10 largely contributes to its nuclear transport upon LTP-inducing stimuli. Thus, RNF10 accumulates in the nucleus following long-distance protein transport.

We next used a validated chemical LTP (cLTP) protocol (*Otmakhov et al., 2004*; *Oh et al., 2006*), which is known to activate synaptic NMDARs in primary hippocampal neurons. As expected, induction of cLTP leads to an increase of CREB-Ser133 (P-CREB; *Bito et al., 1996*) and GluA1-Ser845 (P845-GluA1; *Marcello et al., 2013*) phosphorylation (data not shown). Under these experimental conditions, cLTP induction produced a significant decrease in endogenous RNF10 immunostaining along dendrites (*Figure 8D*) and a concomitant significant increase of RNF10 localization in the nucleus as revealed by western blotting in a crude nuclear fraction (*Figure 8G*) and co-localization assays with the nuclear marker Dapi (*Figure 8F* and *Figure 9B*). Consistent with the above-described results (see *Figure 7H*), cLTP detached RNF10 from GluN2A (*Figure 8E*). These effects were not correlated to an alteration of RNF10 total protein abundance as determined by WB analysis performed with neuronal lysates (data not shown; +9.5% ± 8.7%, cLTP vs. control; p>0.05, n=8). Finally, we used anisomycin to exclude the possibility that de novo RNF10 protein synthesis could account for the nuclear accumulation of RNF10 following cLTP (*Figure 8F,G*). Anisomycin treatment did not prevent nuclear accumulation of RNF10 in this assay (*Figure 8F,G*).

The RNF10 (1–611) protein fragment, lacking the NLS2 motif, fails to localize to the nucleus (*Figure 7A*). Taking this into account, we induced cLTP in the presence of an exogenously added RNF10 NLS2 peptide, thus competing with endogenous RNF10/importin binding. RNF10 NLS2 peptide is mostly composed of basic residues having a very high similarity with the most common cell-permeable peptides (CPPs) such as the TAT peptide (*Farkhani et al., 2014*). However, before the evaluation of the effect of NL2 peptide on RNF10 subcellular distribution, we first verified the ability of this peptide to act as a CPP, that is, that it can cross the plasma membrane and enter neurons. As shown in *Figure 9A*, we detected TideFluor™-tagged TAT and NLS2 peptides into neurons to a similar extent, thus indicating the capability of the NLS2 peptide to work as a CPP. Notably, NLS2 peptide induced a significant reduction of RNF10/importin α1 interaction when compared to treatment with TAT (used as control peptide; *Figure 9B*). The NLS2 peptide completely prevented the LTP-dependent increase in nuclear RNF10, as indicated by immunofluorescence (*Figure 9C,D*) and by immunoblot analyses of RNF10 protein levels in a P1 crude nuclear fraction (*Figure 9G,H*). Importantly, treatment with NLS2 peptide also prevented the decrease in dendritic RNF10 labeling observed after induction of cLTP (*Figure 9E,F*). Overall, these results further confirm that NLS2-dependent binding to importin represents a key step for RNF10 trafficking to the nucleus.

Conversely, induction of chemical long-term depression (cLTD) did not modify RNF10 binding to GluN2A (*Figure 9I*; +0.9% ± 4.8%, cLTD vs. control; p>0.05) and the nuclear distribution of RNF10 in hippocampal neurons (*Figure 9J*).

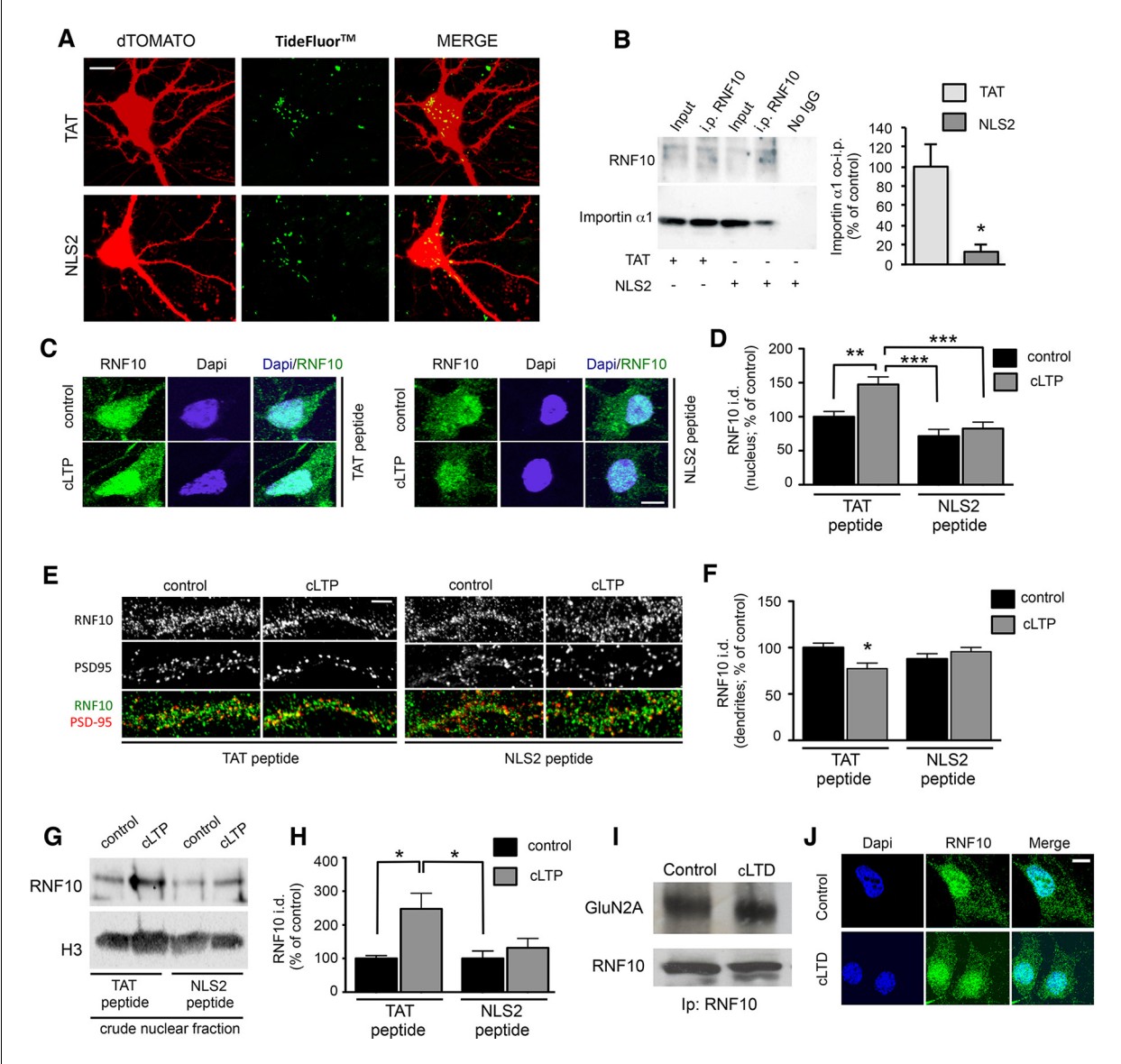

**Figure 9.** NLS2 peptide but not anysomicin blocks RNF10 accumulation in the nucleus after induction of cLTP. (A) Confocal images from living hippocampal neurons transfected with dTomato (*DIV10*) and treated with TAT-TF2 or NLS2-TF2 coniugated peptide (*DIV14*). Samples were illuminated with 543 nm and 488 nm to visualize respectively neurons and peptide. The representative image shows the presence of peptide (green) within the neuron (red, DTOMATO), demonstrating the capability of crossing the plasmatic membrane; scale bar: 10 μm. (B) Representative co-i.p. assay performed by using anti-RNF10 antibody and showing the interaction between RNF10 and importin α1 in primary hippocampal neurons (*DIV14*) treated with NLS2 peptide (active) or TAT (control) peptide. No IgG lane: control lane in absence of the antibody. WB analysis was performed with importin α1 and RNF10 antibodies. The histogram shows the quantification of importin α1 interaction with RNF10 expressed as % of control (TAT; n=3; *p<0.05; unpaired Student's t-test). (C, E) Hippocampal neurons (*DIV14*) were treated with NLS2 peptide (active) or TAT (inactive) peptide for 24 hr and then cLTP was induced in the presence of the same peptides. Confocal images show the immunolabeling for RNF10 (green) and the staining for Dapi (blue) in the nucleus (C) or PSD-95 (red) along dendrites (E); scale bars: 10 μm (C) and 4 μm (E). (D, F) The histograms show the quantification of RNF10 signal in the nucleus (D) and along dendrites (F) after the induction of cLTP in the presence of TAT or NLS2 peptides expressed as % of control [n=10,11 (D), n=17–19 (F)]; *p<0.05; **p<0.01; *p<0.001; one-way ANOVA, followed by Tukey post-hoc test). (G, H) WB analysis for RNF10 from P1 crude nuclear fraction purified from hippocampal neurons after the induction of cLTP in the presence of NLS2 or TAT peptide (G). The histogram (H) shows the quantification of RNF10 integrated density normalized on Histone-H3 and expressed as % of control (n=4, *p<0.05; one-way ANOVA, followed by Tukey post-hoc test). (I) Representative co-i.p. assay performed with an antibody against RNF10 from cell lysates of hippocampal neurons following induction of cLTD. WB analysis was performed using antibody for RNF10 and GluN2A. (J) Confocal images of the soma of hippocampal neurons (*DIV14*) treated with a protocol to induce cLTD and then immunolabeled for RNF10 (green) and stained with Dapi (blue). Scale bar: 10 μm.

## RNF10 translocation to the nucleus regulates the expression of specific target genes

We next asked whether RNF10 translocation from dendritic spines to the nucleus produced a modulation of the protein level of Meox2/RNF10 target genes. First, we analyzed the expression of p21[WAF1/cip1], a known Meox2 target gene (*Lin et al., 2005*; *Malik et al., 2013*). Bic treatment increased p21[WAF1/cip1] levels as evaluated by immunofluorescence (*Figure 10A,B*) and Western blot analysis (*Figure 10C*). Up-regulation of p21[WAF1/cip1] was significantly attenuated by co-treatment with the NMDAR blocker MK801 (*Figure 10A–C*). Similarly, stimulation of synaptic NMDARs induced an increase in p21[WAF1/cip1] that was blocked by bath application of MK801 (*Figure 10D*). Notably, the increase in p21[WAF1/cip1] protein levels following stimulation of synaptic NMDARs was completely prevented by RNF10 silencing (*Figure 10E*), indicating that modulation of p21[WAF1/cip1] expression is RNF10-dependent. In addition, the same effect on p21[WAF1/cip1] levels was observed following induction of cLTP. Again, the increased expression of p21[WAF1/cip1] was prevented by RNF10 silencing (*Figure 10F*). Finally, RNF10 silencing decreased the expression of the newly identified RNF10 target gene Ophn1 (see *Figure 4G,H*) also after synaptic stimulation (*Figure 10G*).

Altogether these results indicate that stimulation of synaptic NMDARs or induction of LTP leads to RNF10 translocation to the nucleus, where it binds to Meox2 and stimulates the expression of RNF10/Meox2 target genes (*Lin et al., 2005*; *Malik et al., 2013*), such as p21[WAF1/cip1] and Ophn1.

## Down-regulation of RNF10 expression prevents cLTP expression

We finally analyzed whether RNF10 silencing leads not only to a decrease of dendritic spine density (*Figure 4A–E*) but also to an alteration of the plastic properties of synapses. To this end, we recorded miniature EPSCs (mEPSCs) in hippocampal neurons before and for 2 hr after cLTP induction. In basal conditions, down-regulation of RNF10 led to a significant decrease of the frequency (*Figure 11A,C*) but not the amplitude (*Figure 11A,B*) of mEPSCs. Notably, no alteration of mEPSCs frequency was observed by co-expressing shRNF10 with the wild-type human variant of RNF10 resistant to shRNA (FlagRNF10; *Figure 11A,C*). Furthermore, while in control neurons cLTP induction resulted in a long-lasting increase of the amplitude and frequency of mEPSCs, this effect was totally prevented in neurons transfected with shRNF10 (*Figure 11D,E*). Again, co-transfection of FlagRNF10 fully rescued a physiological long-lasting increase of the amplitude (*Figure 11D*) and frequency (*Figure 11E*) of mEPSCs.

We then tested whether different levels of RNF10 expression could affect not only spine density (see *Figure 4A–E*) but also interfere LTP-dependent modulation of dendritic spine size (*Bosch and Hayashi, 2012*). To this end, we overexpressed or silenced RNF10 in hippocampal neurons (*DIV7*) followed by induction of cLTP (*DIV14*). As expected, cLTP increased significantly PSD-95 cluster width in control neurons (*Figure 11F,G*). RNF10 overexpression produced an increase in PSD-95 clusters width also in absence of stimulation and occluding any further increase of PSD-95 clusters width following induction of cLTP (*Figure 11F*). On the other hand, RNF10 silencing blocked any cLTP-dependent modification of PSD-95 clusters (*Figure 11G*). Evaluation of GluA1 synaptic cluster size confirmed that RNF10 silencing precludes any increase in GluA1 clusters width induced by cLTP (*Figure 11I*).

## Discussion

This report describes a novel *synapse-to-nucleus* signaling pathway that specifically links activation of synaptic GluN2A-containing NMDARs to nuclear gene expression. We demonstrate that RNF10 is highly expressed in the nucleus as well as at synapses, where it is part of the NMDAR complex and directly interacts with the cytoplasmic tail of the GluN2A subunit of NMDARs. RNF10 dissociates from the NMDAR complex in an activity-dependent manner and we provide compelling evidence for the importin-dependent long-distance transport from synapto-dendritic compartments to the nucleus.

The intracellular C-terminal domain of GluN2A contains several target sequences for downstream signaling and scaffolding molecules (*Sanz-Clemente et al., 2013*; *Sun et al., 2016*). Among others, RNF10 and CaM share the same GluN2A binding region (aa 991–1029; *Bajaj et al., 2014*). Our results indicate a preferential formation of RNF10/GluN2A complex in resting conditions that is

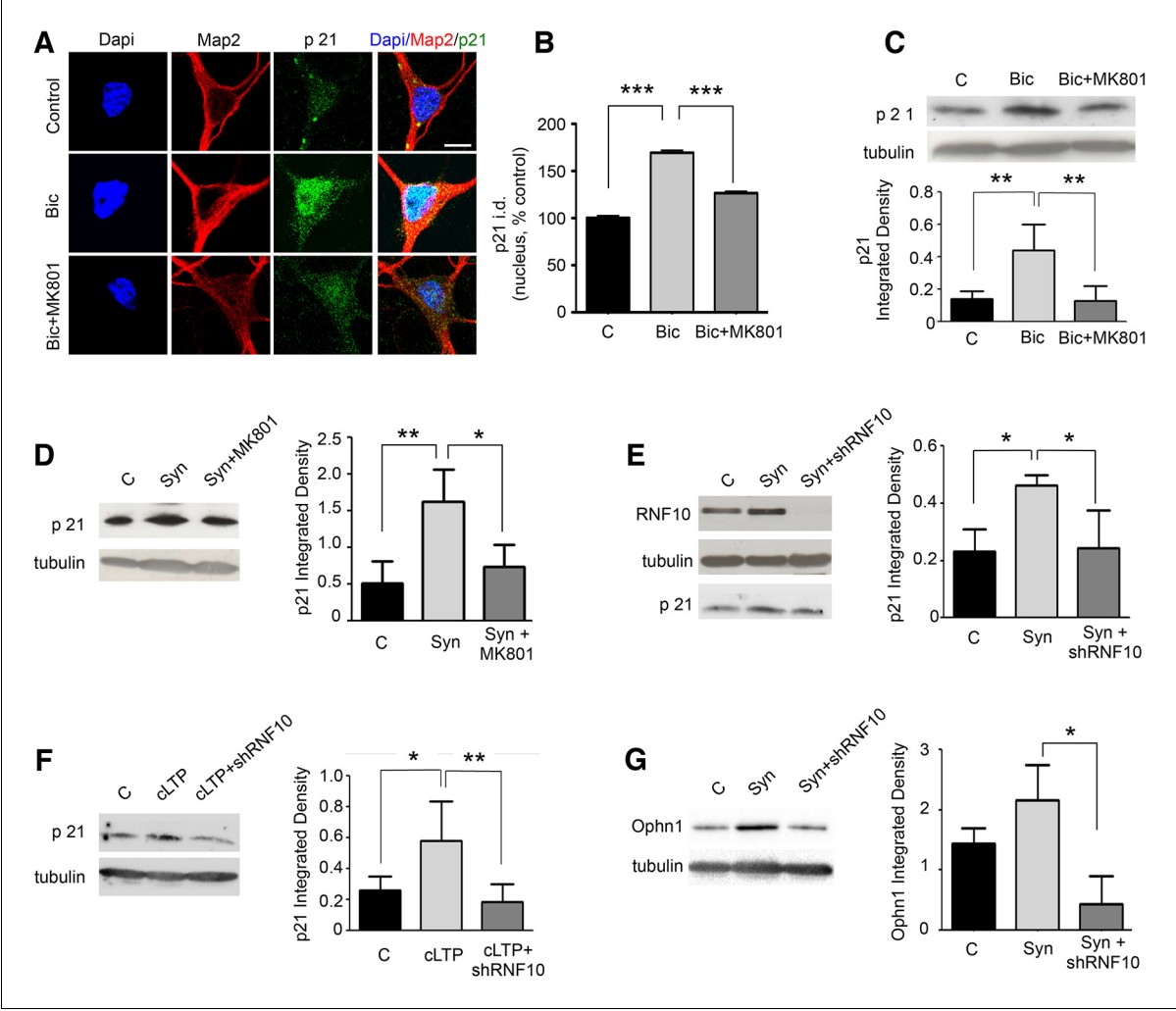

**Figure 10.** RNF10 nuclear trafficking regulates the expression level of specific target genes. (**A**) Hippocampal neurons (*DIV14*) were treated with Bic in the presence or absence of MK801, immunolabeled for MAP2 (red), p21$^{WAF1/cip1}$ (green) and stained with Dapi (blue); scale bar: 10 μm. (**B**) Histogram showing the quantification of p21$^{WAF1/cip1}$ signal in the nucleus expressed as % of control (n=15, ***p<0.001 Bic vs control and Bic+MK801 vs Bic; one-way ANOVA, followed by Bonferroni post-hoc test). (**C**) WB analysis from cell lysates of hippocampal neurons treated with Bic in presence or absence of MK801. The histogram shows the quantification of p21$^{WAF1/cip1}$ integrated density normalized on tubulin (n=5, **p<0.01 Bic vs control and Bic+MK801 vs Bic; one-way ANOVA, followed by Bonferroni post-hoc test). (**D**) WB analysis for p21$^{WAF1/cip1}$ and tubulin performed on cell lysates from *DIV14* hippocampal neurons treated for synaptic stimulation (Syn; see Materials and methods) in the presence or absence of MK801. The histogram shows the quantification of p21$^{WAF1/cip1}$ levels normalized on tubulin (expressed as integrated density) (n=5, **p<0.01, Syn vs control; *p<0.05 Syn+MK801 vs Syn; one-way ANOVA, followed by Bonferroni post-hoc test). (**E**) WB analysis for RNF10, p21$^{WAF1/cip1}$ and tubulin performed on cell lysates from hippocampal neurons virally infected with pLKO-shRNF10 or control vector and treated at *DIV14* for Syn. The histogram shows the effect of RNF10 silencing on p21$^{WAF1/cip1}$ expression levels normalized on tubulin (n=4, *p<0.05, Syn vs control and Syn+shRNF10 vs Syn; one-way ANOVA, followed by Bonferroni post-hoc test). (**F**) WB analysis for p21$^{WAF1/cip1}$ and tubulin from hippocampal extracts following induction of cLTP in the presence or absence of viral infection with pLKO-shRNF10. The graph shows the modulation of p21$^{WAF1/cip1}$ levels (following normalization on tubulin) (n=5, *p<0.05, cLTP vs control, **p<0.01, cLTP + pLKO-shRNF10 vs cLTP; one-way ANOVA, followed by Bonferroni post-hoc test). (**G**) WB analysis for Ophn1 and tubulin performed on cell lysates from hippocampal neurons virally infected with pLKO-shRNF10 or control vector and treated at *DIV14* for Syn. The histogram shows the effect of RNF10 silencing on Ophn1 expression levels normalized on tubulin (n=3, *p<0.05, Syn+shRNF10 vs Syn; one-way ANOVA, followed by Bonferroni post-hoc test).

disrupted in favor of CaM/GluN2A complex following synaptic activity-dependent calcium influx. Overall, our data show that RNF10 binding to GluN2A plays a key role for RNF10 anchoring at the excitatory synapse but also interfere with the formation of CaM/NMDAR complex. Moreover, the presence of RNF10 in close proximity to the receptor channel allows for a rapid calcium-dependent

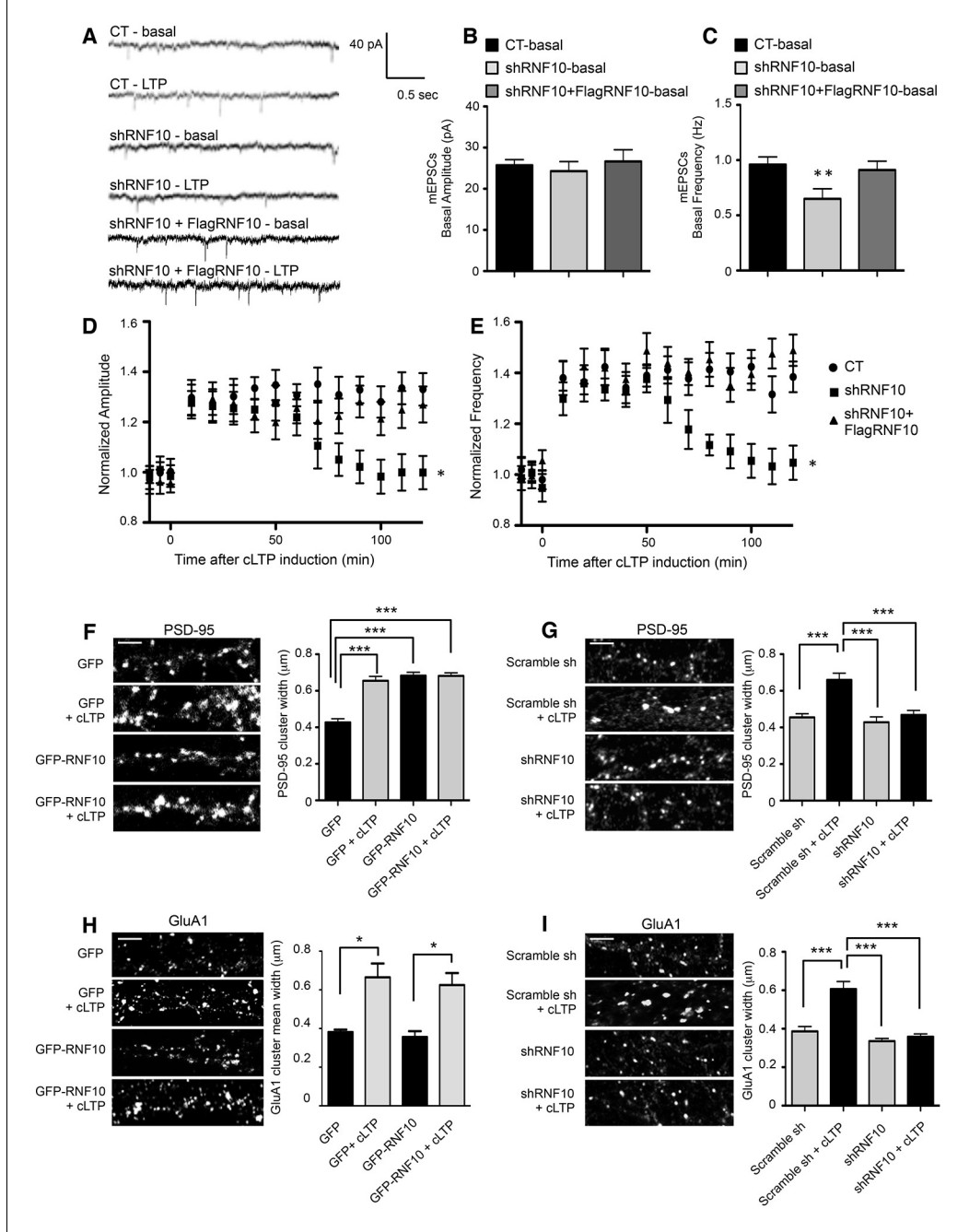

**Figure 11.** Down-regulation of RNF10 expression prevents cLTP expression. (**A**) mEPSCs recorded at -60 mV before (basal), and after cLTP induction in hippocampal neurons transfected or not (CT) with shRNF10 or with shRNF10 + FlagRNF10. (**B, C**) Basal amplitude (**B**) and frequency (**C**) of mEPSCs, prior cLTP induction (**p<0.01, shRNF10 vs CT). (**D, E**) Time-course of amplitude (**D**) and frequency (**E**) of mEPSC following cLTP induction (*p<0.05, shRNF10 vs CT). (**F, H**) Effects of RNF10 overexpression on cLTP-induced modifications of dendritic spine morphology. Confocal images of hippocampal neuron dendrites (*DIV14*) transfected at *DIV7* with GFP or GFP-RNF10, treated for cLTP induction and immunolabeled for PSD-95 (**F**) or GluA1 (**H**); scale bar: 4 μm. The histograms show the quantification of PSD-95 (**F**) and GluA1 (**H**) clusters width (PSD-95: n=6, ***p<0.001; GluA1: n=8–10, *p<0.05; one-way ANOVA, followed by Bonferroni post-hoc test). (**G, I**) Effects of RNF10 silencing on cLTP-induced modifications of dendritic spine morphology. Confocal images of hippocampal neuron dendrites (*DIV14*) transfected at *DIV7* with scramble sh or shRNF10, treated for cLTP induction and immunolabeled for PSD-95 (**G**) or GluA1 (**I**); scale bar: 4 μm. The histograms show the quantification of PSD-95 (**G**) and GluA1 (**I**) clusters width (PSD-95: n=6, ***p<0.001; GluA1: n=8–10, ***p<0.001; one-way ANOVA, followed by Bonferroni post-hoc test).

dissociation of RNF10/GluN2A complex following NMDAR activation and for the subsequent formation of RNF10/importin complex.

Here, we show that interaction of RNF10 with importin α1 represents a key step for its nuclear translocation. However, we also describe that RNF10 can interact with the majority of importin α isoforms including the importin α5 previously described to be associated with the NMDAR complex (*Jeffrey et al., 2009*). Taken together, these findings suggest that synaptonuclear trafficking of RNF10 is involved in the control of gene expression, which is necessary for LTP-type synaptic plasticity in hippocampal neurons. Very few studies are available addressing the role of RNF10 in the CNS, and they mainly address its association with the transcription factor Meox2, which regulates cell proliferation and differentiation (*Seki et al., 2000*; *Lin et al., 2005*; *Hoshikawa et al., 2008*; *Malik et al., 2013*). A very recent study performed in P19 carcinoma cell line and in mouse cerebellar granule cells suggests that RNF10 acts as a positive regulator of neuronal differentiation, as its knockdown reduced the number of cells expressing early and late neuronal markers (*Malik et al., 2013*). Notably, microarray, subsequent real-time PCR and western blotting analysis allowed us to find, among others, novel RNF10 target genes known to be involved in the regulation of the excitatory synapse function and morphology (*Vogt et al., 2007*; *Michaluk et al., 2011*; *Bagni et al., 2012*; *Ramakers et al., 2012*; *Pavlowsky et al., 2012*). RNF10 silencing induced a dramatic up-regulation of MMP9 expression and a concomitant reduction of Arhgef6, ArhGap4 and Ophn1 levels. Intriguingly, all these genes are mutated or dysregulated in intellectual disability syndromes in humans and/or in the corresponding mouse models (*Vogt et al., 2007*; *Michaluk et al., 2011*; *Ramakers et al., 2012*), which are characterized by various alterations in dendritic spines, thus suggesting a key role in synaptic effects observed following RNF10 silencing. In this view, modulation of RNF10 expression in hippocampal neurons demonstrated its relevant role in regulating dendritic spine morphology under resting conditions as well as following activity-dependent plasticity. In particular, RNF10 silencing leads to a significant decrease in spine density that is fully rescued by co-transfection of shRNA-resistant RNF10 construct. As a direct consequence of this dramatic morphological event, we also observe a concomitant reduction in the expression of proteins almost exclusively localized at the excitatory PSD, such as PSD-95, AMPARs GluA1 subunit and NMDARs GluN2A subunit. Notably, overexpression of the RNF10(1–611) construct unable to translocate to nucleus leads to a significant reduction in PSD-95 positive puncta along dendrites. This result strongly indicates that an impaired RNF10-mediated synaptonuclear signaling and not a reduction of RNF10 synaptic levels is responsible for the reduction of dendritic spine density observed following RNF10 silencing.

The present and other studies indicate that NMDAR complex is likely a very rich source of protein messengers that are capable of trafficking to the nucleus. Most important, the present study provides convincing evidence that different NMDAR signals will indeed induce the nuclear import of different messengers. Several reports have documented that NMDARs activation leads to a differential modulation of nuclear gene expression, depending upon their localization (*Hardingham and Bading, 2010*). In general, activation of synaptic NMDARs promotes the expression of synaptic plasticity-related genes, whereas activation of extrasynaptic NMDARs has been correlated to CNS pathological conditions (*Hardingham and Bading, 2010*). The protein messenger Jacob, following long-distance transport and nuclear import, can encode and transduce the synaptic and extrasynaptic origin of GluN2B NMDAR signals to the nucleus and might dock a NMDA-receptor-derived signalosome to nuclear target sites in a stimulus-dependent manner (*Karpova et al., 2013*). Notably, here we show that activation of synaptic but not extrasynaptic NMDARs leads to an increase in RNF10 nuclear localization. Similarly to what was previously described for Jacob (*Behnisch et al., 2011*), we observe that RNF10 moves to the nucleus after induction of LTP but not LTD, modulating the expression of Meox2 target genes. However, in contrast to Jacob, inhibition of GluN2A- but not GluN2B-containing NMDARs is sufficient to block RNF10 trafficking to the nucleus, its interaction with the transcription factor Meox2 and the expression of the Meox2 target gene $p21^{WAF1/cip1}$. Another intriguing finding includes the observation that the transcription factor CREB2 also known as ATF4 transits to the nucleus only after induction of NMDAR-dependent cLTD but not cLTP (*Lai et al., 2008*). Transcriptional regulators at synaptic sites like CRTC1 can also be phosphorylated by synaptic signals in a complex manner and translocation of CRTC1 to the nucleus might link specific types of stimuli to specific changes in gene expression.

Overall our data together with the findings from other recent reports (*Kaushik et al., 2014*; *Rishal and Fainzilber, 2014*; *Panayotis et al., 2015*), suggest that *synapse-to-nucleus* protein transport provide specific ways to inform the nucleus about very different types of synaptic activity. In this context, RNF10 represents a novel synaptonuclear protein messenger responsible for long-lasting re-shaping of dendritic spines as observed after specific synaptic stimuli and required for postsynaptic modifications needed to convey LTP induction.

## Materials and methods

### Purification of post-synaptic densities, triton insoluble postsynaptic fractions and crude nuclear fractions

Post-synaptic densities (PSDs) from rat hippocampus were isolated as previously reported (*Gardoni et al., 1998*). To purify the postsynaptic Triton-insoluble fraction (TIF), cell lysates were centrifuged at 13,000 g for 15 min at 4°C. The resulting pellet was resuspended in 150 mM KCl, 0.5% Triton and spun at 100000 g for 1 hr at 4°C. The final pellet (TIF) was homogenized with a glass-glass potter in 20 mM Hepes buffer containing Complete. All purifications were performed in the presence of complete sets of protease and phosphatase inhibitors (Roche Diagnostics, Monza, Italy) Protein content of the samples has been quantified by using Bio-Rad (Hercules, CA, USA) protein assay. After measuring protein concentration, all samples have been standardized at 1 µg/µl concentration and the same protein amount loaded in each lane for western blot analysis.

### Organotypic hippocampal slice, neuronal cultures, transfection and treatments

Organotypic hippocampal slice cultures were prepared as previously described (*Pellegrini Giampietro et al., 1999*). Hippocampal neuronal primary cultures were prepared from embryonic day 18–19 (E18-E19) rat hippocampi as previously described (*Piccoli et al., 2007*). Neurons were transfected at *DIV7* using calcium-phosphate method. For induction of 'Bic' treatment: hippocampal neurons (*DIV14*) or organotypic hippocampal slices (*DIV14*) were incubated with Bicuculline (50 µM; Tocris) and 2.5 mM 4-Aminopyridine (4-AP;Tocris) in Neurobasal medium supplemented with B27; 'Syn' treatment: stimulation of synaptic NMDA receptors was obtained by treating hippocampal neurons at *DIV14* with 50 µM Bicuculline (Tocris), 2.5 mM 4-AP and 5 µM ifenprodil in Neurobasal medium supplemented with B27; 'Extrasyn#1' treatment: stimulation of extrasynaptic NMDA receptors was obtained by pretreating hippocampal neurons at *DIV14* with 50 µM Bicuculline, 2.5 mM 4-AP and 10 µM MK801 for 30 min, washing them with Neurobasal medium supplemented with B27, incubating with 50mM KCl (Carlo Erba) for 10 min and then with 40 µM Glutamate (Sigma); 'ExtraSyn#2' treatment was performed as previously described by *Karpova and co-workers (2013)*. cLTP (*Otmakhov et al., 2004*; *Oh et al., 2006*): Primary neuronal cultures at *DIV14* were first incubated in ACSF (125 mM NaCl, 2.5 mM KCl, 1 mM MgCl$_2$, 2 mM CaCl$_2$, 33 mM D-glucose and 25 mM HEPES, pH 7.3) for 30 min, followed by 16 min of stimulation with 0.05 mM Forskolin (Sigma Aldrich), 0.1 mM Picrotoxin (Tocris) and 100 nM Rolipram (Calbiochem) in ACSF (no MgCl$_2$) to induce NMDAR-dependent cLTP. Control was incubated in ACSF. After stimulation, neurons were replaced in regular ACSF for 2 hr. cLTD (*Marcello et al., 2013*): primary neuronal cultures at *DIV14* were first incubated in ACSF, followed by incubation in ACSF (in presence of MgCl$_2$) and 50 µM NMDA (Sigma Aldrich) for 10 min. Neurons were then replaced in regular ACSF for 1 hr to induce cLTD. For treatment with 'NLS2 peptide' hippocampal neurons (*DIV14*) were treated with active (NLS2: RKRKRQKQK) or control (TAT: GRKKRRQRRRPQ) peptide for 24 hr and then treated to induce cLTP the presence of the same peptides.

### COS-7 and HEK293T experiments

COS-7 cells were maintained in DMEM + Glutamax medium (GIBCO-BRL) supplemented with 10% Fetal Bovine Serum (FBS) (GIBCO-BRL) and Pen/Strep (GIBCO-BRL). The day before transfection, COS-7 cells were placed in a 6 wells multiwell (for cells lysis) or 12 wells multiwell (for cells immunostaining), then cells were transfected using lipofectamine method (Invitrogen). After 36 hr, COS-7 cells were lysed for co-immunoprecipitation and western blotting or fixed for immunostaining.

RNF10-tagged with GFP was co-expressed with importin α isoforms (*KPNA1-KPNA6*) tagged with tagRFP in HEK293T cells. Thirty-six hours post-transfection proteins were extracted with RIPA buffer containing phosStop (50 mM Tris pH=8, 1% NP-40, 0.5% NO deoxycholate, 0.1% SDS, 150 mM NaCl, 1% Triton X-100, protease inhibitor). RNF10-GFP was immunoprecipitated from cell extract using anti-GFP MicroBeads (MACSMolecular). Co-immunoprecipitated importinerase-α isoforms were detected in complex with RNF10 using anti-tagRFP antibodies (Evrogen).

## Gene expression and microarray

Real-time quantitative PCR (qPCR) was performed as previously described (*De Fabiani et al., 2003*). The following gene expression assay kits from Life Technologies have been used: Rn01409258_g1 for S100a11; Rn01504461_g1 for Rac2; Rn01526492_m1 for Apbb1ip; Rn01422083_m1 for Il18; Rn01430875_g1 for Timp1; Mm00623991_m1 for Itgb8; Rn01513693_m1 for Thbs1; Rn00580728_m1 for Cd36; Rn00579162_m1 for Mmp9. For microarray experiments, RNA was analyzed by the Genopolis Consortium using an Affimetrix platform (GEO accession number GSE69267). Data was primarily analyzed using the TAC and Partek Genomic suite softwares. The Robust Multichip Average (RMA) method was employed to calculate probe set intensity (*Irizarry et al., 2003*). Differentially-expressed genes whose fold change was higher or equal than 1.75 (up-regulated genes) and lower or equal to 0.65 (down-regulated genes) with a p value lower than 0.05 were selected as significantly modulated. The expression value of each probe is divided by the mean value of the scramble controls and the log2 of this ratio has been reported. Gene Ontology biological process and enriched pathways analyses were performed using Panther software (http://www.pantherdb.org/).

## Co-immunoprecipitation assays (co-i.p.)

Hippocampi from adult rats were homogenized at 4°C in an ice-cold buffer containing 0.32 M Sucrose, 1 mM Hepes, 1 mM NaF, 0.1 mM PMSF, 1 mM MgCl in presence of protease inhibitors (Complete, GE Healthcare, Mannheim, Germany) and phosphatase inhibitors (PhosSTOP, Roche Diagnostics GmbH, Mannheim, Germany), using a glass-Teflon homogenizer. Homogenates were then centrifuged at 1000 g for 5 min at 4°C, to remove nuclear contamination and white matter. The supernatant was collected and centrifuged at 13,000 g for 15 min at 4°C. The resulting pellet was resuspended in hypotonic buffer (1 mM Hepes with Complete) and referred as P2 crude membrane fraction. Aliquots of proteins from homogenate, P2 crude membrane fractions or PSDs were incubated overnight at 4°C in RIPA buffer containing 50mM Tris HCl (pH 7.2), 150 mM NaCl, 1% NP-40, 0.5% deoxycholic acid, 0.1% sodium dodecyl sulphate (SDS) in a final volume of 150 µl with the antibody. Pre-incubation with 1% SDS was used for co-i.p. experiments performed in PSDs (*Gardoni et al., 2001*). As a control, one sample was incubated in the absence of the antibody (no IgG lane) and another sample in presence of an irrelevant antibody. Protein A-sepharose beads (Sigma-Aldrich) were added and incubation was continued for 2 hr, at room temperature, with shaking. Beads were collected by centrifugation and washed three times with RIPA buffer before adding sample buffer for SDS-PAGE and boiling for 5 min. Beads were collected by centrifugation, all supernatants were applied onto 7%–12% SDS-PAGE and revealed by western blotting.

## In situ proximity ligation assay (PLA)

PLA was performed as previously described (*Söderberg et al., 2006*; *Augusto et al., 2013*) in primary hippocampal neurons. Cells were fixed with 4% PFA - 4% Sucrose for 5 min at 4°C. Cells were then rinsed in PBS, permeabilized with 0.1% Triton X-100 in PBS for 15 min at room temperature and blocked with 5% BSA in PBS for 30 min at room temperature. Subsequently, cells were incubated with goat polyclonal anti-RNF10 antibody (1:200; Santa-Cruz) and rabbit polyclonal anti-GluN2A antibody (1:200; Lifetechnologies) overnight at 4°C. After washing in PBS, the cells were incubated for 1 hr at 37°C with the PLA secondary probes anti-goat Plus and anti-rabbit Minus (Olink Bioscience). Afterward, the cells were washed twice with Duolink II Wash Buffer A (Olink Bioscience) and incubated with the ligation-ligase solution (Olink Bioscience) for 30 min at 37°C. After a new rinse, the cells were incubated with DNA polymerase (1:80; Olink Bioscience) in the amplification solution (Olink Bioscience) for 100 min at 37°C. After washes in consecutive decreasing concentrations with Duolink II Wash Buffer B (Olink Bioscience), the cells were then incubated with chicken

polyclonal anti-GFP/rabbit polyclonal anti-MAP2 (1:300/1:400; Milipore/Milipore) overnight at 4°C. After washing in PBS, the cells were incubated for 1 hr at room temperature with secondary goat anti-chicken-AlexaFluor 488/goat anti-rabbit-AlexaFluor 647 (Lifetechnologies). The cells were washed with PBS and mounted on slides with Fluoroshield mounting medium (Sigma).

### Cloning, expression, and purification of glutathione S-transferase (GST) fusion protein

The C-terminal tail of the GluN2A subunit was subcloned downstream of glutathione S-transferase (GST) in the *BaMHI* and *HindIII* sites of the expression plasmid pGEX-KG by PCR using the Pfu polymerase (Stratagene) on a GluN2A cDNA template (kind gift from S. Nakanishi). GST-GluN2A fusion proteins containing the cytoplasmic domain of GluN2A (839–1464 or 1049–1464 or 1244–1461 or 1244–1464) were expressed in *Escherichia coli* and purified on glutathione agarose beads (Sigma Aldrich, St. Louis, MO) as previously described (*Gardoni et al., 2001*).

### Pull-down assay

Three hundred micrograms of proteins were incubated with GST to a final volume of 1 ml with Tris Buffered Saline solution for 45 min with shaking. Samples were centrifuged at 1000 rpm for 5 min and the pellets were discarded. Supernatants were incubated for 2 hr with GST fusion proteins or GST alone. Beads were washed four times with TBS and 0.1% Triton X-100. Bound proteins were resolved by SDS-PAGE and subjected to immunoblot analysis.

### Confocal studies

For co-localization and morphological studies, treated hippocampal neurons were fixed 7 min in 4% paraformaldehyde plus 4% sucrose in PBS at room temperature; then extensively washed with PBS supplemented with $CaCl_2$ and $MgCl_2$ (PBS-C.M), permeabilized with 0.2% Triton X100 in PSB-C.M and blocked for 2 hr at room temperature with 5% BSA in PBS-C.M. Primary and secondary antibodies were applied in 5% BSA in PBS-C.M. Cells were chosen randomly for quantification from different coverslips. Fluorescence images were acquired by using Zeiss Confocal LSM510 Meta system with a sequential acquisition setting at 1024x1024 pixels resolution; for each image two up to four 0.5μm sections were acquired and a z projection was obtained (*Malinverno et al., 2010*). All images were acquired standing the signals within the linear range in order to perform a reliable quantification (avoiding the presence of any saturated pixel) and to compare appropriately all experimental conditions. For analysis of dendritic spine morphology, primary hippocampal neurons were transfected with GFP-containing constructs at *DIV7*. Cells were fixed and immunolabeled for GFP at *DIV14*. The SP5 CLSM system (Leica-Microsystems, Mannheim, Germany) equipped with Diode (405 nm), Argon (488 nm) and Diode Pumped Solid State (561 nm) lasers was used for time-lapse imaging of RNF10 fused to tdEOS. Images were taken with HCX APO L20x/1.00W objective (Leica, Germany). Along the z-axis at list 10 optical sections with focus depth of 300–400 nm were taken in order to cover the complete volume of imaged neurons. Quick change 18 mm Chamber with Field Stimulation (RC-49MFS, Warner Instruments) was used for mounting coverslips with living overexpressing RFN10-tdEOS neurons in Neurobasal medium (Gibco) on the microscope stage. Master-8/U 8-channel programmable pulse generator (Science Products) was used to generate timing and two channels were synchronized to produce complex patterns of pulses (1 ms pulse duration) with inter pulse interval of 20 ms. To eliminate the distortion and thereby to achieve a rectangular pulse permitting accurate measurement, the constant-current stimulus isolation unit (SIU-102, Warner Instruments) was used in the output of a stimulus generator. The LTP was induced by applying current pulses 18s@50 Hz via field stimulation electrode. Response of cultured hippocampal neurons on field synaptic stimulation was verified by $Ca^{2+}$-imaging with Fluo-4-AM (1 μM) (data not shown).

### Electrophysiology

Miniature EPSCs were recorded on *DIV14* hippocampal neurons at room temperature, at a holding potential of -60 mV. The recording pipettes had resistance of around 5 MΩ when filled with the following medium (in mM): 140 CsCl, 0.5 $CaCl_2$, 20 EGTA, 10 HEPES, 10 D-glucose, pH 7.2 and osmolarity of 300 mOsm. The high concentration of EGTA avoided slow $Ca^{2+}$-dependent desensitization of NMDARs. Neurons were perfused continuously with the following external medium (in mM): 140

NaCl, 2 CaCl$_2$, 3 KCl, 10 HEPES, 10 D-glucose, 0.01 glycine, 0.01 bicucullin, 0.0003 tetrodotoxin, pH 7.4 and osmolarity of 330 mOsm. Currents were recorded through an Axopatch 200B amplifier, filtered at 1 kHz, digitized at 3 kHz and analyzed using the pClamp 10.0 software of Axon Instrument (Molecular Devices, Sunnyvale, CA).

## Quantification and statistical analysis

Acquisition and quantification of western blotting was performed by means of computer-assisted imaging (ChemiDoc system and Image lab 4.0 software; Bio-Rad). Western blotting experiments have been normalized on tubulin or H3 levels, depending if they have been performed from total cell lysate (homogenate) or P1 nuclear fraction, respectively. Data obtained by pull-down assays were normalized on the amount of GST fusion protein. The data from all quantifications were analyzed using Graphpad Prism software and levels and values were expressed as mean ± standard error of the mean (SEM). Co-localization analysis was performed using Zeiss AIM4.2 software. Statistical evaluations were performed by Student $t$ test or Pearson correlations or, as appropriate, by one-way ANOVA followed by Bonferroni's or Tukey's as a post-hoc test. When appropriate, experiments have been performed in blind conditions.

## Antibodies

The following antibodies were used: monoclonal antibody (mAb) anti-α-calcium/calmodulin-dependent kinase II (αCaMKII), polyclonal antibody (pAb) anti-GluN2A, pAb anti-CREB, mAb anti-Myc, pAb anti-GluA1 and pAb anti-p-CREB (Ser-133) were purchased from Millipore (Billenca, MA,USA); mAb anti-Meox2 and pAb anti-Synaptotagmin were purchased from Abcam (Cambridge, MA, USA); mAb anti-GluR2, mAb anti-GFP, mAb anti-GST and anti-PSD-95 were purchased from NeuroMab (Davis, CA); mAb anti-α-Tubulin and mAb anti-Flag were purchased from Sigma-Aldrich (St. Louis, MO); mAb anti-p21 and mAb anti-Rch1 (importin α1, *KPNA2* gene product) were purchased from BD Biosciences (Franklin Lakes, NJ); pAb anti-histone H3 was purchased from Proteintech (Chicago, IL); pAb anti-RNF10 were purchased from ProteinTech and Santacruz; mAb anti-GluN2B and p-GluA1(845) were purchased from Invitrogen (Carlsbad, CA); pAb anti-BDNF was purchased from Genetex (Irvine, CA); mAb anti-Map2 was purchased from Immunological Sciences (Roma, Italy); mAb anti-JL8 was purchased from Clontech (Mountain View, CA); pAb anti-ArhGap4 was purchased from Novus Biologicals (Cambridge, UK); mAb anti-GFAP, pAb anti-Ophn1 and pAb anti-ArhGef6 were purchased from Cell Signaling (Danvers, MA). Peroxidase-conjugated secondary anti-mouse Ab was purchased from Pierce (Rockford, IL) while peroxidase-conjugated secondary anti-rabbit Ab was purchased from Bio-Rad (Hercules, CA). AlexaFluor secondary Abs were purchased from Invitrogen (Carlsbad, CA).

The specificity of the RNF10 antibody (ProteinTech) in recognizing RNF10 in western blotting and immunocytochemistry was carefully verified by RNF10 overexpression and silencing in primary hippocampal neurons (data not shown). The specificity of the importin α1 antibody (BD Biosciences) in recognizing *KPNA2* gene product in western blotting was carefully verified by *KPNA1-6* overexpression in HEK293T cells (data not shown).

## shRNA

The RNAi Consortium (TRC) shRNA target gene set in pLKO.1 lentiviral vector for rat RNF10 (accession number NM_001011904.1) along with scrambled vector were purchased from Open Biosystems (Thermo Scientific, Milano, Italy). The set contained three vectors (the TRC IDs: TRCN0000041128, mature antisense TTCAGGTTGATCTTCTTAGGG; TRCN0000041131, mature antisense TTTCTGTGAATTGGAGCGACG; and TRCN0000041132, mature antisense ATACCCAGCAATCTCTACCAC), all of them were validated by real-time PCR on rat hippocampal primary cultures. Vector TRCN0000041128 showed the highest level of RNF10 downregulation (>90%, data not shown) and was used in all further experiments (designated as shRNF10). To knock down rat GluN2A gene expression, a set of 4 short hairpin RNA (shRNA) expression vectors in pGFP-V-RS was purchased from Origene (Rockville, MD). The sequences were as follows: A-gacagcactcctatgataacattctggac; B-tctaccagcaggactggtcacagaacaac; C-tgcgagccagatgacaaccacctcagcat; D-agtagaggtggctgtcagcactgaatcca. A scrambled control from Origene (Rockville, MD) was also used as a negative control. Sequence called B showed the best knockdown and was therefore used for further experiments.

For the testing and validation of sh-resistant form of RNF10, rat primary hippocampal cultures were transfected with GFP (Ctrl), rat-shRNF10, RFP-humRNF10, and rat-shRNF10 + TRP-humRNF10. Total RNA was extracted with Tryzol 48 hr after transfection, retrotranscribed (Improm II, Promega) and analyzed by quantitative PCR using primers that specifically amplify rat RNF10 or human RNF10. Transfection with rat-shRNF10 resulted in three fold decrease of endogenous RNF10 mRNA, whereas transfection with human RFP tagged RNF10 resulted in 45-fold increase in mRNA for human RNF10, the level which was not affected by rat-shRNF10 co-transfection. Interestingly, the overexpression of human RNF10 in rat hippocampal cells abolished effect of shRNF10 on levels of endogenous RNF10 mRNA. The specificity of rat-shRNF10-mediated knock down was further tested using human neuroblastoma line SY-S5Y5. No effect on both endogenous and everexpressed RNF10 was observed after transfection with rat-shRNF10.

## Viral constructs

Cloning of pLV-RNF10-EGFP. The cloning of this construct was accomplished in two steps. First, EGFP was amplified from the pRRLSIN.cPPT.hCMV-EGFP.WPRE (pLV-EGFP) plasmid to create in frame sequence with XbaI-SalI sites at 5' of EGFP and XhoI sites at 3' of EGFP using primers: forward 5'-aagcttTCTAGAgaattcGTCGACATGGTGAGCAAGGGCGAGGA; reverse 5'-aagcttCTCGAGTTACTTGTACAGCTCGTCCAT. Amplificate was digested with XbaI and XhoI enzymes and ligated in XbaI-SalI digested pLV vector creating the intermediate pLV-(XS)EGFP bearing XbaI-SalI cassette at 5' of EGFP. After the sequence was confirmed by sequencing, human RNF10 was excised from pCMV-Tag4-RNF10 plasmid by XbaI and SalI and ligated in XbaI/SalI digested pLV-(XS)EGFP yielding pLV-RNF10-EGFP plasmid.

## Production of viral particles

For production of lentiviral particles 2 mln of HEK293T producing cells were plated on 100 mm plates (Costar, Milan, Italy) in complete medium (DMEM, Cat. D5671; 2 mM Glutamine; 100 units/ml penicillin and 100 µg/ml streptomycin (Sigma); 10% Foetal Bovine Serum (FBS, Immunological Sciences, Milano, Italy). For overexpression of RNF10, the day after plating the cells were transfected with 3µg pL-shRNF10 or pGIPZ-scr and with 2.5 µg psPAX2 and 0.75 µg pMD2.G retroviral packaging constructs using Lipofectamine 2000 (Life Technologies, Milan, Italy) in 7 ml of complete medium. For RNF10 silencing, the cells were transfected with 3 µg pLV-RNF10-EGFP, and with 2 µg pMDLg/pRRE, 0.75 µg pRSV-REV, 11 µg pMD2.G lentiviral packaging constructs. 48 hr after transfection culture medium was collected, filtered through 0.45 µm PVDF filter and precipitated with polyethylene glycol (PEG; 5x PEG solution contained: 200 g PEG, Sigma Cat. 89510, 12 g NaCl in 2 mM Tris pH 7.5) overnight. Precipitates were then collected by centrifugation at 1500 xg for 30 min at 4°C, resuspended in cold HEPES balanced salt solution (HBSS, Sigma), aliquoted and snap frozen. Aliquots were kept at -80°C. Each viral batch was tittered by serial dilution infections of rat hippocampal primary cells by means of real-time PCR (for RNF10 silencing) using primers for rat RNF10 (to evaluate efficacy of downregulation): forward 5'-GAACAAGAGACAGGGCAGGT, reverse 5'-GACAGAGGGGTCAGCAGAAA. For experiments, a viral titer was used yielding > 80% of RNF10 downregulation. Virus for overexpression was tittered by FACS, and a titer yielding > 80% EGFP positive cells was used.

## Yeast two hybrid (Y2H) screening

Y2H was conducted according to the manufacture procedure guidelines using the Mate & Plate™ Library - Mouse Brain (Normalized) (cat # 630488, TakaraBio/Clontech Europe, France). Briefly, the GluN2A(839–1461) C-tail (bait) was cloned in the pGBKT7 plasmid and transformed in the AH109 haploid yeast strain (MATa). This was mated overnight with the Mouse Brain normalized library (prays) cloned in pGADT7 transformed in Y187 haploid yeast strain (MATα). Yeast were plated after 24 on selective plate, allowing only the growth of diploid where a protein interaction between the bait and pray protein occurred (absence of Leucine, Adenine, Tryptophan and Histidine). Then the diploid where tested by a colorimetric assay (α-gal) to avoid the presence of false positive. Plasmids (21 positive clones) were extracted from the yeast and sequenced. Five genes were identified following sequencing, one of those being RNF10.

## Acknowledgements

This work was supported by: FP7-PEOPLE-ITN-2008 #238608 (SYMBAD) to MDL; Fondazione CARI-PLO (grant number 0795-2013) to MDL and AAG; FIRB Accordi di Programma RBAP11HSZS to MDL; Agence Nationale de la Recherche [ANR-13-JSV4-0005-01, SYNcity], Région Languedoc-Roussillon (Chercheur d'Avenir) to JP; PRIN2010-2011 2010AHHP5H to FaG; PRIN2010-2011 2010PWNJXK to MDL and PLC; DFG (SFB779 TPB8 / Kr1879/5-1/6-1) and FP7 MC-ITN NPlast to MRK; CBBS to AK and MRK.

We thank Dr. Richard Huganir (Johns Hopkins University School of Medicine, Baltimore, MD) for the GFP-GluN2A construct and Dr. Toru Ogata (The University of Tokyo, Bunkyo-ku, Tokyo, Japan) for the rat RNF10 cDNA. We want to thank Dr. J Wiedermann and Dr. U Thomas for providing tdEOS construct and Dr. M Mikhaylova for help with Jacob-GluN2B co-i.p. experiments.

## Additional information

### Funding

| Funder | Author |
|---|---|
| Cariplo Foundation, Italy | Monica DiLuca |
| European Commission | Monica DiLuca |
| Italian Ministry of Research and University | Fabrizio Gardoni Monica DiLuca |

The funders had no role in study design, data collection and interpretation, or the decision to submit the work for publication.

### Author contributions

MCD, AK, FGa, Conception and design, Acquisition of data, Analysis and interpretation of data, Drafting or revising the article; FGu, DL, NM, SM, MM, EM, JS, TS, AB, AC, Acquisition of data, Analysis and interpretation of data; AAG, LF, PLC, MDL, Conception and design, Drafting or revising the article; JP, MRK, Conception and design, Analysis and interpretation of data, Drafting or revising the article

### Author ORCIDs

Monica Di Luca, http://orcid.org/0000-0003-2298-615X

### Ethics

Animal experimentation: All procedures were performed in accordance with the current European Law and were approved by the Italian Ministry of Health (as indicated in Dlgs N. 295/2012-A).

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
