## [Decision Letter]

Thank you for submitting your work entitled "Ring finger protein 10 is a novel synaptonuclear messenger encoding activation of NMDA receptors in hippocampus" for consideration by *eLife*. Your article has been favorably evaluated by Gary Westbrook (Senior Editor) and two reviewers, one of whom, Eunjoon Kim, is a member of our Board of Reviewing Editors, and another is Johannes Hall.

The reviewers have discussed the reviews with one another and the Reviewing Editor has drafted this decision to help you prepare a revised submission.

Summary:

This manuscript reports the novel role of ring finger protein 10 (RNF10) in the regulation of NMDAR-dependent signaling between synapses and the nucleus. In support of this idea, the authors show that RNF10 is present at synaptic sites and directly interacts with GluN2A. RNF10 knockdown induces reductions in spine density and some synaptic proteins. Activation of synaptic NMDARs induces synaptonuclear translocation of RNF10 through NLS2-dependent interactions with importin α proteins. In addition, LTP-inducing stimuli also induce synaptonuclear RNF10 translocation, leading to the interaction of RNF10 and a transcriptional regulator Meox2, and regulation of the expression of RNF10 target genes.

These are original findings of high interest. It is especially remarkable that RNF10 knockdown not only reduces spine density but also total GluN2A, GluA1, and PSD-95 protein levels. In the future, it seems worthwhile to explore regulation of the transcription of these proteins by RNF10 in more detail. Also notable is the contrast between RNF10 and Jacob, another synaptonuclear shuttle protein as discovered earlier by some of the authors, which associates with GluN2B rather than GluN2A, with RNF10 shuttling induced by activation of GluN2A- and not GluN2B-containing NMDARs whereas Jacob shuttling requires the opposite. Contrasting the differential behaviors of those two proteins is an especially nice aspect of this manuscript.

However, the below issues should be thoroughly addressed before a final decision can be rendered.

Essential revisions:

1) The authors show in Figure 4 that RNF10 knockdown decreases spine density and levels of certain synaptic proteins. Measuring how RNF10 knockdown affects mEPSCs and mIPSCs would strengthen the conclusions. A rescue condition should be included. Ideally, this experiment should be done using cultured slices or virus-infected brain slices rather than dissociated neurons to increase the relevance of the results to more intact circuits.

2) It is unclear how RNF10 dissociates from GluN2A by NMDAR activation. If any mechanism were involved, would it be also involved in the enhanced interaction between RNF10 and importins/Meox2? These mechanisms should be explored to some extent or discussed.

---

## [Author Response]

Essential revisions:

*1) The authors show in Figure 4 that RNF10 knockdown decreases spine density and levels of certain synaptic proteins. Measuring how RNF10 knockdown affects mEPSCs and mIPSCs would strengthen the conclusions. A rescue condition should be included. Ideally, this experiment should be done using cultured slices or virus-infected brain slices rather than dissociated neurons to increase the relevance of the results to more intact circuits.*

We agree with the reviewers that an evaluation of RNF10 function in more intact circuits is of great relevance. Accordingly, we have received very recently from KOMP Repository the RNF10 ko mice (RNF10^tm1a(KOMP)Wtsi^) for in vivo*/*ex vivo studies that will represent the ideal follow up of the present manuscript.

In order to increase the relevance of results obtained in dissociated neurons we replicated the western blotting analysis shown in Figure 4 also from organotypic hippocampal slices infected with pLKO-shRNF10 lentivirus or scramble sequence as a control. As shown in Figure 4 of the revised version of the manuscript, RNF10 silencing resulted in a significant decrease of GluN2A, PSD-95 and GluA1 protein levels in the total cell homogenate of both pLKO-shRNF10-infected dissociated neurons and organotypic slices. Similarly, we also show that RNF10 silencing lead to a significant reduction of the expression of OphN1 expression in experiments performed both in organotypic slices (Figure 4) and dissociate neurons (Figure 10).

Following the reviewers’ indications, electrophysiological experiments have been repeated with the addition of a rescue condition, namely the co-transfection of a sh-resistant form of RNF10 (Flag-RNF10). As shown in Figure 11 of the revised version of the manuscript, the effect of RNF10 silencing on mEPSCs frequency was fully rescued by co-expressing Flag-RNF10 resistant to shRNA (Figure 11). Furthermore, co-transfection of FlagRNF10 fully rescued a physiological long-lasting increase of the amplitude (Figure 11) and frequency (Figure 11) of mEPSCs as observed following induction of LTP.

2) It is unclear how RNF10 dissociates from GluN2A by NMDAR activation. If any mechanism were involved, would it be also involved in the enhanced interaction between RNF10 and importins/Meox2? These mechanisms should be explored to some extent or discussed.

As shown in Figure 3 and Figure 3 of the revised version of the manuscript we have performed additional experiments to address this important issue. Firstly, we have prepared novel GST-GluN2A fusion proteins to identify more precisely the GluN2A domain involved in the interaction with RNF10. As shown in Figure 3 we found that GluN2A(991-1049) is needed for the binding to RNF10. This GluN2A intracellular domain contains the GluN2A(991-1029) domain that has been recently identified as an atypical high affinity calcium/calmodulin binding site (Bajaj G, et al., 2014, Biochem Biophys Res Commun 444: 588-94). Notably, the formation of GluN2A/calmodulin complex was strictly dependent to the presence of calcium. In agreement with these results, here we show by in vitro pull-down assay (Figure 3) and co-IP from transfected COS-7 cells (Figure 3) that incubation with calcium/calmodulin leads to a complete disruption of the GluN2A/RNF10 complex. Overall, these new experiments indicate the existence of a competition between RNF10 and calmodulin for the binding to GluN2A that can take place following NMDA receptor activation and consequent calcium influx.